# SPREAD: SAMPLING-BASED PARETO FRONT REFINEMENT VIA EFFICIENT ADAPTIVE DIFFUSION

**Sedjro Salomon Hotegni, Sebastian Peitz**
Department of Computer Science, TU Dortmund University
Lamarr Institute for Machine Learning and Artificial Intelligence
`{salomon.hotegni,sebastian.peitz}@tu-dortmund.de`

## ABSTRACT

Developing efficient multi-objective optimization methods to compute the Pareto set of optimal compromises between conflicting objectives remains a key challenge, especially for large-scale and expensive problems. To bridge this gap, we introduce SPREAD, a generative framework based on Denoising Diffusion Probabilistic Models (DDPMs). SPREAD first learns a conditional diffusion process over points sampled from the decision space and then, at each reverse diffusion step, refines candidates via a sampling scheme that uses an adaptive multiple gradient descent-inspired update for fast convergence alongside a Gaussian RBF–based repulsion term for diversity. Empirical results on multi-objective optimization benchmarks, including offline and Bayesian surrogate-based settings, show that SPREAD matches or exceeds leading baselines in efficiency, scalability, and Pareto front coverage. Code is available at `https://github.com/safe-autonomous-systems/moo-spread`.

## 1 INTRODUCTION

Multi-objective optimization (MOO) is fundamental in numerous scientific and engineering disciplines, where decision-makers often face the challenge of optimizing conflicting objectives simultaneously (Rangaiah, 2016; Malakooti, 2014; Zhang et al., 2024b). The primary aim is to identify the Pareto front: a set of non-dominated solutions where improving one objective would deteriorate at least one other. Traditional methods for approximating the Pareto front include evolutionary algorithms (Deb, 2011; Zhou et al., 2011), scalarization techniques (Braun et al., 2015; Hotegni & Peitz, 2025), and multiple-gradient descent (MGD) (Désidéri, 2012; Sener & Koltun, 2018) combined with multi-start techniques (i.e., using random initial guesses to obtain multiple points). However, these approaches may struggle with scalability, especially in high-dimensional or resource-constrained settings (Cheng et al., 2021; Li et al., 2024a). As a workaround, domain-specific MOO algorithms have been developed to leverage domain knowledge to improve efficiency and solution quality in targeted settings (e.g., offline MOO (Yuan et al., 2025b), Bayesian MOO (Daulton et al., 2022), or federated learning (Hartmann et al., 2025)), but at the expense of broader applicability. These challenges underscore the need for MOO methods capable of efficiently adapting to large-scale, high-dimensional, and computationally expensive problem settings. Developing such universal approaches would not only streamline the optimization process, but would also broaden the applicability of MOO techniques across different domains.

Recent advancements in generative modeling have shown promise in addressing complex optimization problems (Garciarena et al., 2018; Yuan et al., 2025a). In particular, diffusion models such as Denoising Diffusion Probabilistic Models (DDPMs), have demonstrated remarkable capabilities in generating high-quality samples across various domains (Ho et al., 2020; Yang et al., 2023). Their iterative refinement process aligns well with the principles of MOO, offering a potential pathway to efficiently approximate the Pareto front. In this work, we introduce **SPREAD** (**S**ampling-based **P**areto front **R**efinement via **E**fficient **A**daptive **D**iffusion), a novel diffusion-driven generative framework designed to tackle multi-objective optimization across diverse problem settings. Our approach leverages the strengths of diffusion models to iteratively generate and refine candidate solutions, guiding them towards Pareto optimality. SPREAD applies a conditional diffusion modeling approach, where a conditional DDPM is trained on points sampled from the input space, allowing

the model to effectively learn the underlying structure of the objective functions and steer the generation process toward promising regions. To further enhance convergence towards Pareto optimality, SPREAD incorporates an adaptive guidance mechanism inspired by the multiple gradient descent algorithm (Désidéri, 2012), dynamically guiding the sampling process to regions likely to contain optimal solutions. Furthermore, to promote diversity among the generated solutions and ensure an approximation of the entire Pareto front, SPREAD utilizes a Gaussian RBF repulsion mechanism (Buhmann, 2000) that discourages clustering, mitigates mode collapse and encourages exploration in the objective space.

We evaluate SPREAD on diverse MOO problems including two challenging, resource-constrained scenarios: offline multi-objective optimization (Xue et al., 2024) and Bayesian multi-objective optimization (Knowles, 2006). In each case, we benchmark our method against state-of-the-art approaches specifically tailored for these settings. Our empirical results demonstrate that SPREAD not only achieves competitive performance but also offers superior scalability and adaptability across different problem domains. Our contributions can be summarized as follows: $(a)$ We propose a novel diffusion-based generative framework for MOO that effectively approximates the Pareto front. $(b)$ We introduce a novel conditioning approach along with an adaptive guidance mechanism inspired by MGD to improve convergence towards Pareto optimal solutions. $(c)$ We implement a diversity-promoting strategy to ensure a comprehensive and well-distributed set of solutions. $(d)$ We validate our approach on challenging MOO tasks, demonstrating its effectiveness and generalizability.

## 2 RELATED WORK

We now situate our approach within prior work on generative modeling for multi-objective optimization, and gradient-based methods relevant to our settings. An extended discussion of related work is provided in Appendix E.

**Generative Modeling for Multi-Objective Optimization**  Recent work explores alternatives to traditional search methods, such as evolutionary algorithms or acquisition-based optimization, by directly generating Pareto optimal candidates. ParetoFlow (Yuan et al., 2025a) uses flow-matching with a multi-objective predictor-guidance module to steer samples toward the front in the offline setting, showing that guided generative samplers can cover non-convex fronts efficiently. In parallel, PGD-MOO (Annadani et al., 2025) trains a dominance-based preference classifier and uses it for diffusion guidance to obtain diverse Pareto optimal designs from data. For Bayesian settings, CDM-PSL (Li et al., 2025a) couples unconditional/conditional diffusion with Pareto set learning to propose candidate points under tight evaluation budgets. Our approach differs by conditioning a diffusion transformer on objectives and applying step-wise, MGD-inspired guidance together with an explicit diversity force, yielding both convergence and spread without a separate preference classifier.

**Gradient-based Methods for Pareto Set Discovery**  A complementary line of research explicitly profiles the Pareto set by moving a population with repulsive interactions or by smoothing scalarizations. PMGDA (Zhang et al., 2025) extends the classical MGDA (Désidéri, 2012) by sampling multiple descent directions in a probabilistic manner, thus improving stability and coverage in high dimensions. Smooth Tchebycheff scalarization (STCH) provides a lightweight differentiable scalarization with favorable guarantees (Lin et al., 2024). Leveraging hypervolume gradients, HV-Grad (Deist et al., 2021) updates solutions toward the Pareto front while preserving diversity, while MOO-SVGD (Liu et al., 2021) employs Stein variational gradients to transport particles and obtain well-spaced fronts. Extending this line of work, SPREAD integrates MGD directions into a DDPM denoising process, using them as adaptive guidance signals within diffusion sampling.

## 3 PRELIMINARIES

To set the stage for our method, we review the fundamental concepts on which our approach is based.

### 3.1 DENOISING DIFFUSION PROBABILISTIC MODELS (DDPMS)

As powerful generative models, Denoising Diffusion Probabilistic Models excel at producing high-quality samples in a wide range of applications, such as image synthesis (Dhariwal & Nichol, 2021), speech generation (Kong et al., 2020), and molecular design (Hoogeboom et al., 2022). These models operate by simulating a forward diffusion process, where Gaussian noise is incrementally added to

data, followed by a learned reverse process that denoises the data step by step. In the conditional setting, DDPMs generate data samples conditioned on auxiliary information $c$, enabling controlled generation aligned with specific attributes or constraints. The forward diffusion process gradually corrupts a data point $\mathbf{x}_0$ over $T$ timesteps:

$$q(\mathbf{x}_t|\mathbf{x}_{t-1}) = \mathcal{N}(\mathbf{x}_t; \sqrt{1-\beta_t}\mathbf{x}_{t-1}, \beta_t \mathbf{I}), \tag{1}$$

where $\beta_t$ is a variance scheduling parameter, often chosen according to a linear (Ho et al., 2020) or a cosine (Nichol & Dhariwal, 2021) schedule. After $T$ steps, $\mathbf{x}_T$ approaches a standard Gaussian distribution. The aim is to reconstruct $\mathbf{x}_0$ from $\mathbf{x}_T$ by learning a parameterized model $\hat{\epsilon}_\theta(\cdot)$ that predicts the added noise at each timestep $t$, conditioned on $c$. The model is trained to minimize the following loss:

$$\mathcal{L}_s(\theta) = \mathbb{E}_{\mathbf{x}_0, \epsilon, t, c}\left[\|\epsilon - \hat{\epsilon}_\theta(\mathbf{x}_t, t, c)\|^2\right], \tag{2}$$

where $\epsilon \sim \mathcal{N}(0, \mathbf{I})$ is the true noise added to $\mathbf{x}_0$ at a randomly chosen timestep $t \in \{1, \ldots, T\}$ to obtain $\mathbf{x}_t$, at each epoch. Specifically, from equation 1 we have after $t$ timesteps $\mathbf{x}_t = \sqrt{\bar{\alpha}_t}\,\mathbf{x}_0 + \sqrt{1-\bar{\alpha}_t}\,\epsilon$, with $\bar{\alpha}_t = \prod_{i=1}^{t}(1-\beta_i)$.

At inference (post-training), sampling starts from pure noise $\mathbf{x}_T \sim \mathcal{N}(0, \mathbf{I})$ and iteratively denoises it using the learned reverse process:

$$\mathbf{x}_{t-1} = \frac{1}{\sqrt{1-\beta_t}}\left(\mathbf{x}_t - \frac{\beta_t}{\sqrt{1-\bar{\alpha}_t}}\hat{\epsilon}_\theta(\mathbf{x}_t, t, c)\right) + \sqrt{\beta_t}\mathbf{z}, \tag{3}$$

where $\mathbf{z} \sim \mathcal{N}(0, \mathbf{I})$. To enhance sample quality and control, guidance techniques can be applied: classifier guidance introduces gradients from a separately trained classifier to steer the generation process (Dhariwal & Nichol, 2021), while classifier-free guidance interpolates between conditional and unconditional predictions within the same model, allowing for flexible control without additional classifiers (Ho & Salimans, 2022).

## 3.2 MULTI-OBJECTIVE OPTIMIZATION (MOO)

Multi-objective optimization involves optimizing multiple conflicting objectives simultaneously (Eichfelder, 2008; Peitz & Hotegni, 2025):

$$\min_{\mathbf{x} \in \mathcal{X}} \mathbf{F}(\mathbf{x}) = (f_1(\mathbf{x}), \ldots, f_m(\mathbf{x})), \tag{MOP}$$

where $\mathcal{X}$ is the decision space, and each $f_j : \mathcal{X} \longrightarrow \mathbb{R}$, $j \in \{1, \ldots, m\}$ represents an objective function. Throughout this paper, we assume that each objective function is continuously differentiable.

**Definition 1** (Pareto Stationarity). *A solution $\mathbf{x}^* \in \mathcal{X}$ is said to be Pareto stationary if there exist nonnegative scalars $\lambda_1, \ldots, \lambda_m$, with $\sum_{j=1}^{m} \lambda_j = 1$, such that $\sum_{j=1}^{m} \lambda_j \nabla f_j(\mathbf{x}^*) = 0$.*

Such points are necessary candidates for Pareto optimality but may include non-optimal solutions.

**Definition 2** (Dominance). *A solution $\mathbf{x}' \in \mathcal{X}$ is said to dominate another solution $\mathbf{x} \in \mathcal{X}$ (denoted $\mathbf{x}' \prec \mathbf{x}$) if:  $f_j(\mathbf{x}') \leq f_j(\mathbf{x})$ for all $j = 1, \ldots, m$, and $\exists i \in \{1, \ldots, m\} \mid f_i(\mathbf{x}') < f_i(\mathbf{x})$.*

**Definition 3** (Pareto Optimality). *A solution $\mathbf{x}^* \in \mathcal{X}$ is called Pareto optimal if there is no $\mathbf{x}' \in \mathcal{X}$ such that $\mathbf{x}' \prec \mathbf{x}^*$. It is called weakly Pareto optimal if there is no $\mathbf{x}' \in \mathcal{X}$ such that $f_j(\mathbf{x}') < f_j(\mathbf{x}^*)$ for all $j = 1, \ldots, m$.*

The set $\mathcal{P}$ of all Pareto optimal solutions is called *Pareto set*, and its image $\mathbf{F}(\mathcal{P}) = \{\mathbf{F}(\mathbf{x}^*) : \mathbf{x}^* \in \mathcal{P}\}$, is known as *Pareto front*. Among the various strategies for solving multi-objective optimization problems, gradient-based techniques are of particular relevance, and in our case, multiple gradient descent serves as the key inspiration for the update mechanism within SPREAD.

## 3.3 MULTIPLE GRADIENT DESCENT (MGD)

Multiple gradient descent is a technique designed to find descent directions that simultaneously improve all objectives in MOO (Désidéri, 2012). Given the gradients $\nabla f_j(\mathbf{x})$ for each objective $f_j$, MGD seeks a convex combination of these gradients that yields a common descent direction at each iteration. This is achieved by solving the following optimization problem:

$$\lambda^* = \arg\min_{\lambda \in \Delta_m} \left\|\sum_{j=1}^{m} \lambda_j \nabla f_j(\mathbf{x})\right\|^2, \tag{4}$$

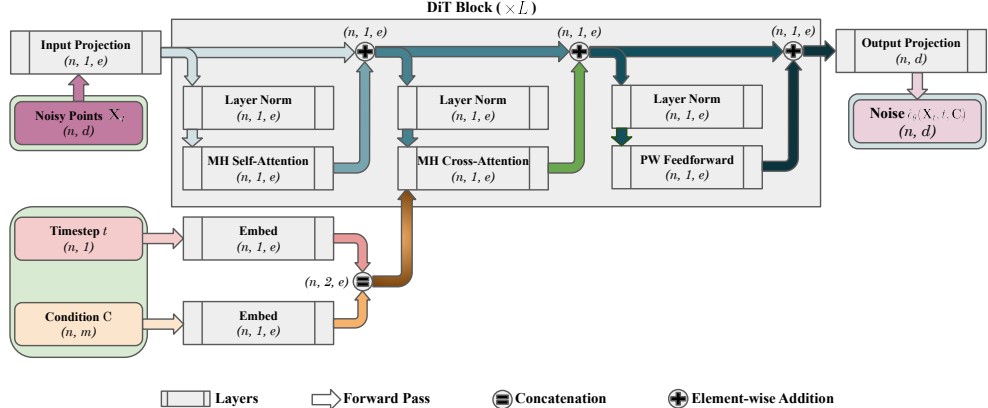

Figure 1: **DiT-MOO architecture.** Diffusion Transformer adapted for multi-objective optimization, where noise prediction is conditioned on objective values condition via multi-head cross-attention.

where $\Delta_m = \{\lambda \in \mathbb{R}^m \mid \sum_{j=1}^m \lambda_j = 1, \lambda_j \geq 0\}$ is the standard simplex. The optimal weights $\lambda^*$ define the aggregated gradient $\mathbf{g}(\mathbf{x}) = \sum_{j=1}^m \lambda_j^* \nabla f_j(\mathbf{x})$, whose negative serves as the common descent direction. The decision variable is then updated using this direction: $\mathbf{x}_{t+1} = \mathbf{x}_t - \eta_t \mathbf{g}(\mathbf{x}_t)$, with $\eta_t$ being the step size at iteration $t$. While MGD ensures convergence to a Pareto stationary point, employing a classical multi-start approach does not inherently promote diversity among solutions. To overcome this drawback, our method incorporates a mechanism that promotes diversity, as detailed in the next section.

## 4 METHOD

In this section, we first present the core components of our method for solving an MOP in an *online setting* (full access to the objective functions), and then discuss how we adapt them to different resource-constrained settings. We adopt a Transformer-based noise-prediction network, DiT-MOO (Fig. 1), adapted from the Diffusion Transformer (DiT) architecture (Peebles & Xie, 2023), for stable and scalable sampling. The model takes as input a batch of $n$ noisy decision variables $\mathbf{X}_t \in \mathbb{R}^{n \times d}$, together with a timestep $t$ and a condition $\mathbf{C}$, and outputs the predicted noise $\hat{\epsilon}_\theta(\mathbf{X}_t, t, \mathbf{C})$. A cosine schedule (Nichol & Dhariwal, 2021) is considered for the variance scheduling parameter $\beta_t$. Further architectural details are provided in Appendix A.4.

**Training** For a given MOP, we sample $N$ points $\{\mathbf{x}^i\}_{i=1}^N = \mathbf{X}$ from the decision space $\mathcal{X} \subseteq \mathbb{R}^d$ via Latin hypercube sampling (McKay et al., 2000) to create the training dataset. Our DiT-MOO is then trained using the loss $\mathcal{L}_s$ (equation 2), on pairs $(\mathbf{x}^i, \mathbf{c}^i)$ with

$$\mathbf{c}^i = \mathbf{F}(\mathbf{x}^i) + \Xi, \quad \Xi \in (0, \infty)^m. \tag{5}$$

During sampling, however, we condition on the original objective vector $\mathbf{F}(\mathbf{x}^i)$. The shift $\Xi$ can be any vector with strictly positive entries, fixed for the entire dataset or varying per point or batch. The following theorem establishes the key advantage of this conditioning approach.

**Theorem 1** (Objective Improvement). *Let $\mathbf{X} \subset \mathcal{X}$ be a training dataset with distribution $P_\mathbf{X}$. Let $\Xi \in (0, \infty)^m$, independent of $\mathbf{X}$, and define the training label*

$$\mathbf{C} := \mathbf{F}(\mathbf{X}) + \Xi. \tag{6}$$

*For a conditioning value $\mathbf{c}$ in the support of $\mathbf{C}$, denote by $P_{\mathbf{X}|\mathbf{C}=\mathbf{c}}$ the true conditional data distribution and by $Q_\theta(\cdot \mid \mathbf{c})$ the distribution produced by a conditional DDPM when sampling conditioned on $\mathbf{c}$. Assume the sampler approximates the true conditional training distribution in total-variation TV distance by at most $\tau \in [0, 1)$:*

$$\mathrm{TV}\big(Q_\theta(\cdot \mid \mathbf{c}), P_{\mathbf{X}|\mathbf{C}=\mathbf{c}}\big) = \sup_A |Q_\theta(A \mid \mathbf{c}) - P_{\mathbf{X}|\mathbf{C}=\mathbf{c}}(A)| \leq \tau. \tag{7}$$

*Fix any initialization $\mathbf{x}_T \in \mathcal{X}$ and set $\mathbf{c} := \mathbf{c}_T = \mathbf{F}(\mathbf{x}_T)$. If $\mathbf{c}_T$ lies in the support of $\mathbf{C}$, and we draw $\mathbf{x}_0 \sim Q_\theta(\cdot \mid \mathbf{c}_T)$, then:*

$$\mathbb{P}(\mathbf{x}_0 \prec \mathbf{x}_T) \geq 1 - \tau. \tag{8}$$

*In other words, conditioning the reverse diffusion on $\mathbf{F}(\mathbf{x}_T)$ yields, with probability at least $1 - \tau$, a sample that dominates $\mathbf{x}_T$.*

The proof of this theorem is provided in Appendix A.1.

**Sampling**   Let $\mathbf{X}_T = \{\mathbf{x}_T^i\}_{i=1}^n \subset \mathcal{X}$ denote $n$ random initial points. We refine them by iteratively applying the reverse diffusion step (equation 3), augmented with a guided update that (i) aligns each sample with its MGD direction and (ii) encourages dispersion in the objective space to promote diversity. Specifically, this guidance is implemented via an additive term, balancing objective improvement (in the spirit of Section 3.3) with spreading along the Pareto front, together with a small noise term. At each sampling step $t$, the update is therefore:

$$\mathbf{X}_t' \longleftarrow \frac{1}{\sqrt{1 - \beta_t}}\Big(\mathbf{X}_t - \frac{\beta_t}{\sqrt{1 - \bar{\alpha}_t}}\,\hat{\epsilon}_\theta(\mathbf{X}_t, t, \mathbf{C})\Big) + \sqrt{\beta_t}\,\mathbf{z}$$
$$\mathbf{X}_{t-1} \longleftarrow \mathbf{X}_t' - \eta_t\,\tilde{\mathbf{h}}_t(\mathbf{X}_t') \tag{9}$$

where the condition $\mathbf{C}$ is the batch of the objective values related to $\mathbf{X}_t$, and

$$\tilde{\mathbf{h}}_t(\mathbf{X}_t') = (\tilde{\mathbf{h}}_t^i)_{i=1}^n = (\mathbf{h}_t^i)_{i=1}^n + \gamma_t^{\mathrm{T}}\delta_t, \tag{10}$$

are the guidance directions. Here, $\delta_t \in \mathbb{R}^d$ is a random perturbation added to the main directions $(\mathbf{h}_t^i)_{i=1}^n$, and $\gamma_t = (\gamma_t^1, \ldots, \gamma_t^n)^{\mathrm{T}} \in \mathbb{R}^n$ are scaling parameters that control the strength of this perturbation. We choose $\mathbf{h}_t^i$, $i = 1, \ldots, n$ to balance two objectives:

(i) *Alignment with the MGD directions:* Let $\mathbf{g}_t^i = \mathbf{g}(\mathbf{x}_t'^i)$, $i = 1, \ldots, n$ be obtained as defined in Section 3.3. The main directions are chosen to maximize the average inner product

$$\frac{1}{n}\sum_{i=1}^n \langle \mathbf{g}_t^i, \mathbf{h}_t^i \rangle. \tag{11}$$

(ii) *Diversity in objective space:* Define $(\mathbf{y}_t^i)_{i=1}^n = \mathbf{Y}_t = \mathbf{F}\big(\mathbf{X}_t' - \eta_t\,((\mathbf{h}_t^i)_{i=1}^n + \gamma_t^{\mathrm{T}}\delta_t)\big)$. The main directions are chosen so as to minimize the Gaussian RBF repulsion function (Buhmann, 2000)

$$\Gamma_t(\mathbf{Y}_t) = \frac{2}{n(n-1)}\sum_{1 \le i < j \le n} \exp\Big(-\frac{\|\mathbf{y}_t^i - \mathbf{y}_t^j\|^2}{2\,\sigma^2}\Big), \tag{12}$$

where $\sigma > 0$ is the length-scale.

Balancing the alignment objective (equation 11) with the diversity requirement (equation 12), we obtain the main directions by solving the following sub-problem:

$$(\mathbf{h}_t^i)_{i=1}^n = \arg\min_{(\mathbf{u}^i)_{i=1}^n}\Big\{ -\frac{1}{n}\sum_{i=1}^n \langle \mathbf{g}_t^i, \mathbf{u}^i \rangle + \nu_t\,\Gamma_t\Big(\mathbf{F}\big(\mathbf{X}_t' - \eta_t\,((\mathbf{u}^i)_{i=1}^n + \gamma_t^{\mathrm{T}}\delta_t)\big)\Big)\Big\}, \tag{13}$$

where $\nu_t \ge 0$. In practice, we solve this sub-problem by performing a fixed number of gradient descent steps, which provides an approximation of the main directions while keeping the computational cost manageable. In the case where $\nu_t = 0$, the main directions $\mathbf{h}_t^i$, $i = 1, \ldots, n$, are well aligned with the MGD directions and thus inherit their descent properties. This assumption leads to the following theorem:

**Theorem 2.** *Assume each objective function $f_j$ is continuously differentiable, and that $\nu_t = 0$ for all $t \in \{1, \ldots, T\}$. Let, at reverse timestep $t$,*

$$a_{i,j} = \big\langle \nabla f_j(\mathbf{x}_t'^i), \mathbf{h}_t^i \big\rangle, \qquad b_{i,j} = \big\langle \nabla f_j(\mathbf{x}_t'^i), \delta_t \big\rangle,$$

*with $a_{i,j} > 0$ for all $i = 1, \ldots, n$ and $j = 1, \ldots, m$. Define*

$$\gamma_t^i = \begin{cases} \rho\,\min_{j:\,b_{i,j}<0}\Big(-\dfrac{a_{i,j}}{b_{i,j}}\Big),\ 0 < \rho < 1, & \text{if any } b_{i,j} < 0, \\ \zeta,\quad \zeta > 0, & \text{otherwise,} \end{cases} \tag{14}$$

*where $\rho$ controls the magnitude of the scaling parameters $\gamma_t^i$, and $\zeta$ denotes an arbitrary positive scalar. Then, $-\tilde{\mathbf{h}}_t^i = -\big(\mathbf{h}_t^i + \gamma_t^i\delta_t\big)$ serves as a common descent direction for all objectives at $\mathbf{x}_t'^i$.*

**Algorithm 1** SPREAD (Online Setting)

**Input:** DiT-MOO architecture (untrained model), a multi-objective optimization problem (MOP).
**Parameter:** epochs $E$, timesteps $T$, sample size $n$.
**Output:** approximate pareto optimal points $\mathcal{P}_0$.

1: DiT-MOO training via Algorithm 2.
2: Initialize $n$ random points $\mathbf{X}_T = \{\mathbf{x}_T^i\}_{i=1}^n \subset \mathcal{X}$
3: $\mathcal{P}_T \leftarrow \mathbf{X}_T$
4: **for** $t = T$ **to** 1 **do**
5:     $(\mathbf{g}_t^i)_{i=1}^n \leftarrow$ Get the MGD directions via Section 3.3.
6:     $(\mathbf{h}_t^i)_{i=1}^n \leftarrow$ Get the main directions via equation 13.
7:     $(\tilde{\mathbf{h}}_t^i)_{i=1}^n \leftarrow$ Get the guidance directions via equation 10.
8:     $\mathbf{X}_{t-1} \longleftarrow$ Get the denoised points via equation 9.
9:     $\mathcal{P}_{t-1} \leftarrow$ Use crowding distance (Appendix A.5) to get the top-$n$ non-dominated points from $\mathbf{X}_{t-1} \cup \mathcal{P}_t$.
10: **end for**
**Return:** $\mathcal{P}_0$

**Algorithm 2** Training (Online Setting)

**Input:** DiT-MOO as the noise prediction network $\hat{\epsilon}_\theta(\cdot)$, a multi-objective optimization problem (MOP).
**Parameter:** epochs $E$, timesteps $T$.
**Output:** a trained noise prediction network $\hat{\epsilon}_\theta(\cdot)$.

1: Sample $N$ points $\{\mathbf{x}^i\}_{i=1}^N = \mathbf{X} \subset \mathcal{X}$ using Latin hypercube sampling (Appendix A.5).
2: $\{\beta_t\}_{t=1}^T \leftarrow$ Get the variances via a cosine schedule (Appendix A.5).
3: **for** epoch $= 1$ **to** $E$ **do**
4:     $t \leftarrow \text{Uniform}(\{1, \ldots, T\})$
5:     $\mathbf{X}_t \leftarrow \sqrt{\bar{\alpha}_t} \mathbf{X} + \sqrt{1 - \bar{\alpha}_t} \epsilon$, with $\epsilon \sim \mathcal{N}(0, \mathbf{I})$, and $\bar{\alpha}_t \leftarrow \prod_{i=1}^t (1 - \beta_i)$.
6:     $\mathbf{C} \leftarrow \mathbf{F}(\mathbf{X}_t) + \Xi$, with $\Xi \in (0, \infty)^m$ an arbitrary vector with strictly positive entries.
7:     Take gradient descent step on $\nabla_\theta \|\epsilon - \hat{\epsilon}_\theta(\mathbf{X}_t, t, \mathbf{C})\|^2$.
8: **end for**
**Return:** $\hat{\epsilon}_\theta(\cdot)$

We provide the proof of this theorem in Appendix A.2. While $\nu_t = 0$ guarantees a common descent direction for all objectives, it is a very strong and restrictive assumption, since a moderate value of $\nu_t$ is necessary to achieve good coverage of the Pareto front. A discussion on the theory for the general case $\nu_t > 0$ can be found in in Appendix A.3, including the sketch of a proof. An ablation study illustrating this trade-off is presented in Appendix D (Figure 9). To determine the batch $\eta_t$ of step sizes at timestep $t$ (equation 9), we employ an Armijo backtracking line search (Armijo, 1966). This ensures sufficient decrease in the objective functions at each timestep $t$, prevents overly aggressive steps, and adapts to local curvature (Fliege & Svaiter, 2000).

The proposed SPREAD framework for solving multi-objective optimization problems is summarized in Algorithm 1. The final set $\mathcal{P}_0$ of approximate solutions is obtained as the top-$n$ non-dominated points from the union $\mathbf{X}_0 \cup \cdots \cup \mathbf{X}_T$. More specifically, for two successive reverse timesteps $t$ and $t-1$, we define $\mathcal{P}_{t-1}$ as the top-$n$ non-dominated points from the union $\mathbf{X}_{t-1} \cup \mathcal{P}_t$ (with $\mathcal{P}_T = \mathbf{X}_T$ initially), using crowding distance (Deb et al., 2002a) to preserve diversity (preferring non-dominated solutions that are less crowded in objective space).

## 4.1 EXTENSION TOWARDS SURROGATE-BASED OPTIMIZATION

Beyond the classical (online) setting, SPREAD extends naturally to resource-constrained multi-objective optimization, where true objective evaluations are expensive or limited and surrogate models are used. Such challenges arise in domains like offline MOO and Bayesian MOO, which require dedicated multi-objective optimization methods to handle restricted or costly evaluations.

**Offline MOO:** In offline multi-objective optimization, the true objective functions are unavailable. Instead, one relies on a pre-collected dataset $\mathcal{D} = \{(\mathbf{x}, \mathbf{F}(\mathbf{x})), \mathbf{x} \in \mathcal{X}\}$ to train a surrogate function $\tilde{\mathbf{F}}$ which serves as a proxy model for the objectives (Xue et al., 2024). To adapt SPREAD to this setting, we set $\mathbf{X} = \mathcal{D}$ in Algorithm 2, and use $\mathbf{F} = \tilde{\mathbf{F}}$ in Algorithms 1 and 2.

**Bayesian MOO:** A key constraint in multi-objective Bayesian optimization (MOBO) is the limited evaluation budget of an expensive $\mathbf{F}$, which is typically addressed by employing iteratively updated Gaussian process surrogate models. Using simulated binary crossover (SBX) (Deb, 1995) as an auxiliary escape mechanism to avoid local optima, together with the data extraction strategy proposed in CDM-PSL (Li et al., 2025a), we adapt SPREAD to the MOBO setting. The procedure is described in Appendix B (Algorithm 3), along with further details. Moreover, Algorithm 1 from the online setting is adapted to Algorithm 4 using Gaussian processes.

## 5 EXPERIMENTS

### 5.1 ONLINE MOO SETTING

We evaluate our method on a diverse suite of problems, ranging from synthetic benchmarks (ZDT (Zitzler et al., 2000), DTLZ (Deb et al., 2002b)) to real-world engineering design tasks RE (Tanabe & Ishibuchi, 2020). All synthetic problems use an input dimension of $d = 30$. The selected real-world tasks use $d \geq 4$ with continuous decision spaces. The baselines considered are gradient-based MOO methods for Pareto set discovery: PMGDA (Zhang et al., 2025), STCH (Lin et al., 2024), MOO-SVGD (Liu et al., 2021), and HVGrad (Deist et al., 2021). For SPREAD, we train DiT-MOO for 1000 epochs with early stopping after 100 epochs. We set the number of timesteps to $T = 5000$, and each baseline is also run for 5000 iterations. Each method produces a set of 200 points, and the quality of the solutions is assessed using the hypervolume (HV) indicator (Guerreiro et al., 2020). More detailed descriptions of the experimental protocols appear in Appendix C.

Table 1: Hypervolume results averaged over 5 independent runs. The best values are **bold**.

| HV (↑) | $m = 2$ | | | | $m = 3$ | | | | | | $m = 4$ |
|---|---|---|---|---|---|---|---|---|---|---|---|
| **Method** | **ZDT1** | **ZDT2** | **ZDT3** | **RE21** | **DTLZ2** | **DTLZ4** | **DTLZ7** | **RE33** | **RE34** | **RE37** | **RE41** |
| PMGDA | **5.72±0.00** | **6.22±0.00** | 5.85±0.00 | 48.14±0.00 | **22.97±0.00** | 19.69±0.20 | 17.82±0.00 | 43.06±0.00 | 210.07±0.00 | 1.18±0.00 | 901.90±3.36 |
| MOO-SVGD | 5.70±0.00 | 6.21±0.00 | 6.08±0.02 | 20.43±0.32 | 22.61±0.02 | 19.69±0.62 | 13.57±0.03 | 16.26±0.17 | 156.20±0.57 | 1.05±0.09 | 579.53±6.42 |
| STCH | 5.71±0.00 | 5.89±0.00 | 5.44±0.13 | 19.07±0.00 | 22.92±0.01 | 14.55±0.00 | 17.46±0.00 | 12.14±0.00 | 156.72±0.00 | 1.31±0.02 | 506.33±2.86 |
| HVGrad | **5.72±0.00** | **6.22±0.00** | **6.10±0.00** | 43.65±0.00 | 22.93±0.00 | 19.98±0.04 | 17.48±0.05 | 36.13±0.00 | 156.72±0.00 | **1.44±0.00** | 936.17±8.91 |
| **SPREAD** | **5.72±0.00** | **6.22±0.00** | **6.10±0.00** | 70.10±0.01 | 22.91±0.00 | 20.22±0.01 | 18.07±0.01 | 133.76±1.72 | 243.15±0.49 | 1.42±0.00 | **1008.75±6.30** |

Table 2: Results of the $\Delta$-spread diversity measure. The best value, along with those whose mean falls within one standard deviation of it, are shown in **bold**.

| $\Delta$-spread (↓) | $m = 2$ | | | | $m = 3$ | | | | | | $m = 4$ |
|---|---|---|---|---|---|---|---|---|---|---|---|
| **Method** | **ZDT1** | **ZDT2** | **ZDT3** | **RE21** | **DTLZ2** | **DTLZ4** | **DTLZ7** | **RE33** | **RE34** | **RE37** | **RE41** |
| PMGDA | 0.42±0.17 | **0.23±0.01** | 1.57±0.02 | 1.53±0.00 | **0.66±0.02** | 1.71±0.07 | 1.02±0.08 | 1.11±0.00 | 1.46±0.00 | 0.59±0.01 | 1.46±0.01 |
| MOO-SVGD | 0.78±0.20 | 1.16±0.11 | 0.90±0.08 | 1.01±0.00 | 1.31±0.01 | 1.02±0.09 | 0.71±0.03 | 1.00±0.00 | 1.20±0.17 | 0.58±0.07 | 1.13±0.04 |
| STCH | 1.01±0.04 | 1.00±0.00 | 1.05±0.03 | 1.00±0.00 | 1.00±0.04 | 1.00±0.00 | 1.06±0.05 | 1.00±0.00 | 1.00±0.00 | 0.80±0.04 | 1.38±0.02 |
| HVGrad | 0.36±0.05 | 1.07±0.05 | 1.08±0.10 | 1.00±0.00 | 1.18±0.05 | 1.56±0.06 | **0.66±0.03** | 1.00±0.00 | 1.00±0.00 | **0.51±0.01** | 1.00±0.02 |
| **SPREAD** | **0.32±0.01** | 0.29±0.02 | **0.53±0.01** | **0.44±0.02** | 0.93±0.05 | **0.80±0.06** | 0.69±0.05 | **0.97±0.02** | **0.88±0.03** | 0.80±0.01 | **0.92±0.03** |

Table 1 reports hypervolume results for problems with two to four objectives. On the bi-objective synthetic problems ZDT1-3, SPREAD matches the best values, while clearly outperforming the baselines on the real-world task RE21. For three objectives, SPREAD achieves the best results on 4 out of the 6 evaluated problems. On the four-objective problem RE41, it attains the highest hypervolume overall. To assess the diversity of the generated solutions for each method, we evaluate the $\Delta$-spread measure as introduced in Deb et al. (2002a). By convention, $\Delta$-spread is set to $+\infty$ when the solutions collapse to a single point. As reported in Table 2, our method yields more diverse solutions on most problems. These results indicate that SPREAD maintains superior performance as the number of objectives increases, providing superior coverage and diversity of the Pareto front in both synthetic and engineering benchmarks. In Appendix D (Figure 7), we show the approximate Pareto optimal points produced by the different methods for four synthetic and four real-world problems.

**Scalability Analysis** We further investigate the scalability of all methods by comparing their computational time as the number $m$ of objectives increases (ZDT1 with $m = 2$, DTLZ2 with $m = 3$, and RE41 with $m = 4$) and as the number $n$ of required samples grows (DTLZ4 with $n = 200, 400, 600, 800$). Unlike the baselines, SPREAD requires a training phase, so we account for both training and sampling times to ensure a fair comparison. As shown in Figure 2(a) and Figure 2(b), PMGDA exhibits the largest growth rate in computational time with increasing $m$ and $n$. In contrast, SPREAD achieves substantially lower computational time than PMGDA, while being moderately more costly than MOO-SVGD, HVGrad, and STH. However, as shown in Figure 2(c) and Figure 2(d), SPREAD consistently offers superior performance in hypervolume and $\Delta$-spread compared to the other methods. Therefore, our method provides a favorable trade-off between efficiency and performance.

**Ablation Study** We present in Table 3 an ablation study on the diversity-promoting mechanisms in SPREAD. Specifically, we evaluate three variants: SPREAD(w/o diversity), with $(\tilde{\mathbf{h}}_t^i)_{i=1}^n = (\mathbf{g}_t^i)_{i=1}^n$; SPREAD(w/o repulsion), with $(\tilde{\mathbf{h}}_t^i)_{i=1}^n = (\mathbf{g}_t^i)_{i=1}^n + \gamma_t^{\mathrm{T}} \delta_t$; and SPREAD(w/o perturbation), with $(\tilde{\mathbf{h}}_t^i)_{i=1}^n = (\mathbf{h}_t^i)_{i=1}^n$. The results indicate that SPREAD(w/o diversity) and SPREAD(w/o repulsion) tend to collapse the solutions to a single point ($\Delta$-spread $= +\infty$). Ignoring the perturbation (SPREAD(w/o perturbation)) has a milder impact on solution quality for some problems. However,

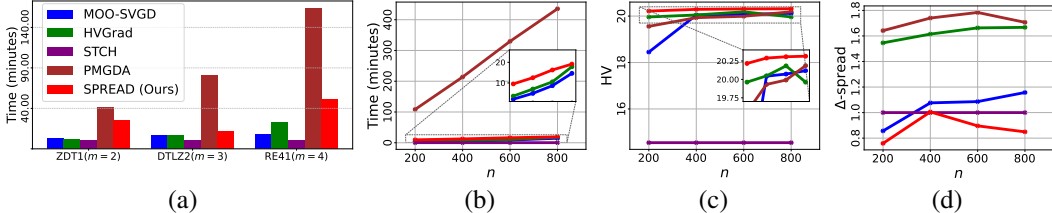

Figure 2: **Scalability.** Comparison of (a) computational time as the number of objectives increases (ZDT1 with $m = 2$, DTLZ2 with $m = 3$, and RE41 with $m = 4$), and (b–d) computational time, hypervolume, and $\Delta$-spread, respectively, as the number of required samples increases (DTLZ4).

Table 3: **Ablation study** on the diversity-promoting mechanisms in SPREAD. Best values are highlighted in **bold**. For $\Delta$-spread, any mean value within one standard deviation of the best is also shown in **bold**. Worst values are shown in red, while best values are shown in blue (HV) and green ($\Delta$-spread).

| Problem | SPREAD | | SPREAD (w/o diversity) | | SPREAD (w/o perturbation) | | SPREAD (w/o repulsion) | |
|---|---|---|---|---|---|---|---|---|
| | HV | $\Delta$-spread | HV | $\Delta$-spread | HV | $\Delta$-spread | HV | $\Delta$-spread |
| ZDT1 | **5.72±0.00** | **0.32±0.01** | 5.06±0.00 | $+\infty$ | **5.72±0.00** | **0.32±0.02** | 4.25±0.08 | 0.88±0.05 |
| ZDT2 | **6.22±0.00** | **0.29±0.02** | 5.89±0.00 | $+\infty$ | **6.22±0.00** | **0.28±0.02** | 4.40±0.14 | $+\infty$ |
| ZDT3 | **6.10±0.00** | 0.53±0.01 | 5.06±0.00 | 0.66±0.00 | **6.10±0.00** | **0.51±0.01** | 4.34±0.07 | 0.84±0.05 |
| RE21 | **70.10±0.01** | 0.44±0.02 | 70.03±0.01 | **0.41±0.02** | 69.01±0.14 | 0.84±0.05 | 70.03±0.03 | 0.51±0.03 |
| DTLZ2 | 22.91±0.00 | 0.93±0.05 | **22.94±0.00** | **0.73±0.03** | 22.8±0.01 | 0.91±0.07 | 22.79±0.04 | 1.06±0.08 |
| DTLZ4 | 20.22±0.01 | **0.80±0.06** | **20.36±0.01** | 0.89±0.11 | 20.01±0.02 | 0.89±0.2 | 20.34±0.02 | 0.97±0.15 |
| DTLZ7 | **18.07±0.01** | **0.69±0.05** | 16.7±0.00 | $+\infty$ | 18.05±0.01 | 0.80±0.03 | 12.84±0.13 | 0.87±0.04 |
| RE33 | **133.76±1.72** | **0.97±0.02** | 8.72±0.65 | **0.99±0.00** | 125.06±0.46 | **0.97±0.04** | 99.89±9.7 | 1.04±0.17 |
| RE34 | **243.15±0.49** | 0.88±0.03 | 236.86±0.94 | 0.97±0.03 | 242.47±0.22 | 0.99±0.02 | 237.34±0.77 | **0.82±0.05** |
| RE37 | **1.42±0.00** | 0.80±0.01 | 1.32±0.00 | 0.98±0.03 | **1.42±0.00** | **0.75±0.03** | 1.40±0.00 | 0.79±0.05 |
| RE41 | 1008.75±6.3 | 0.92±0.03 | 950.45±7.32 | **0.81±0.10** | 969.43±6.44 | 0.93±0.03 | 1011.03±7.52 | **0.78±0.06** |

to maintain a good balance between convergence (HV) and Pareto front coverage ($\Delta$-spread), all diversity-promoting mechanisms of SPREAD are important. The diversity gain observed with SPREAD(w/o diversity) on some problems shows that the stochasticity inherent in DDPM sampling (injected in $\mathbf{X}'_t$ (equation 9)) contributes to the overall diversity of SPREAD. Ablation studies on additional hyperparameters of SPREAD, including $\nu_t$, the perturbation scaling factor $\rho$, and the number of blocks $L$, are provided in Appendix D.

## 5.2 OFFLINE MOO SETTING

In the offline setting, we conduct our evaluation using Off-MOO-Bench (Xue et al., 2024), a unified collection of offline multi-objective optimization benchmarks. Each task is associated with a dataset $\mathcal{D}$ and an evaluation oracle $\mathbf{F}$. During optimization, $\mathbf{F}$ remains inaccessible and is only used to compute the hypervolume of the final solutions. The baselines comprise DNN-based approaches that employ either Multiple Models (MM) or Multi-Head Models (MH), in conjunction with gradient-based algorithms (GradNorm (Chen et al., 2018), and PcGrad (Yu et al., 2020)) or model-based optimization methods (COM (Trabucco et al., 2021), IOM (Qi et al., 2022), ICT (Yuan et al., 2023), RoMA (Yu et al., 2021), and TriMentoring (Chen et al., 2023)) to refine candidate solutions. Additionally, we evaluate the ability of the evolutionary algorithms NSGA-III and MOEA/D to solve offline MOO tasks in Appendix D (Tables 16 and 17). The most relevant baselines for our approach are the generative methods ParetoFlow (Yuan et al., 2025b) and PGD-MOO (Annadani et al., 2025). ParetoFlow utilizes flow-matching models, while PGD-MOO employs a preference-guided diffusion technique. Each algorithm is run with five different random seeds, producing 256 solutions per seed. We evaluate two task groups, Synthetic and RE, with 12

Table 4: **Offline MOO.** Average rank results ($\downarrow$) per task group. Within each group, the overall best method is shown in **bold**, and the best generative approach is highlighted in light gray .

| Method | Synthetic | RE |
|---|---|---|
| $\mathcal{D}$(best) | 9.08 | 11.83 |
| MM | 5.92 | 3.92 |
| MM-COM | 8.00 | 7.42 |
| MM-IOM | 5.67 | 4.33 |
| MM-ICT | 6.50 | 3.83 |
| MM-RoMA | 6.08 | 7.75 |
| MM-TriMentoring | 8.17 | 4.58 |
| MH | 7.58 | 5.67 |
| MH-PcGrad | 5.75 | 6.92 |
| MH-GradNorm | 9.58 | 11.50 |
| ParetoFlow | 7.50 | 6.83 |
| PGD-MOO | 4.58 | 8.75 |
| **SPREAD** | **3.50** | **1.83** |

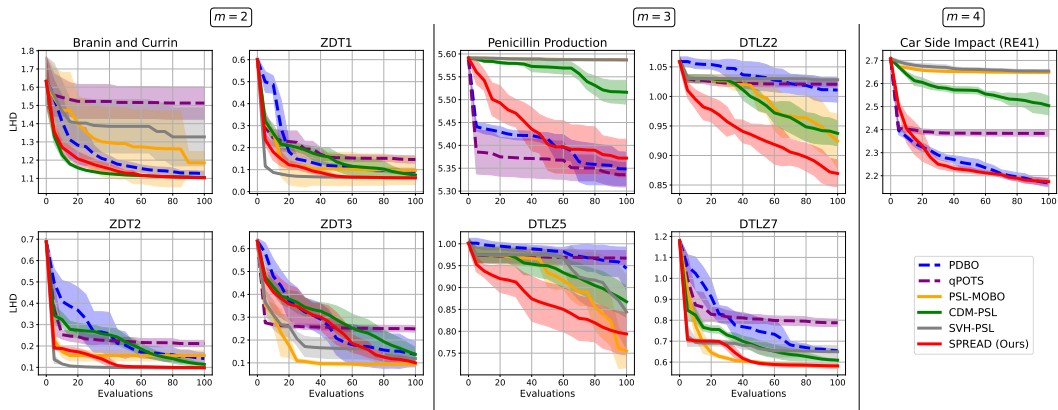

Figure 3: **Bayesian MOO.** Log-hypervolume difference (LHD) over 20 post-initialization steps (totaling 100 function evaluations) on nine MOBO benchmarks: Branin and Currin, ZDT1, ZDT2, ZDT3, Penicillin Production, DTLZ2, DTLZ5, DTLZ7, and Car Side Impact (RE41).

problems in each. Following Xue et al. (2024), we rank algorithms within each task group with respect to their hypervolumes, and use the resulting average rank (↓) as our primary comparison metric. The average rank results are reported in Table 4, while the individual hypervolume results are provided in Appendix D (Tables 14 and 15). Here, "$\mathcal{D}$(best)" denotes the dataset's non-dominated points, serving as a simple baseline. Our method achieves the best average rank across both the synthetic (3.50) and real-world (1.83) task groups, and it outperforms the other generative approaches on most problems in terms of hypervolume (see Tables 14 and 15). These results show that SPREAD effectively leverages static datasets to generate high-quality approximate Pareto fronts without any online queries, matching or even surpassing the performance of state-of-the-art offline multi-objective optimization techniques.

### 5.3 BAYESIAN MOO SETTING

We compare our method against three groups of baselines in multi-objective Bayesian optimization: (1) Pareto set learning–based methods (PSL-MOBO (Lin et al., 2022), SVH-PSL (Nguyen et al., 2025)), (2) acquisition-based methods (PDBO (Ahmadianshalchi et al., 2024), qPOTS (Renganathan & Carlson, 2023)), and (3) CDM-PSL (Li et al., 2025a), a diffusion-based generative approach. We consider nine MOBO problems with 2 or 3 objectives, including the real-world RE41 problem (Car Side Impact), which has 4 objectives All methods were initialized with 100 solutions and then run for 20 iterations, selecting 5 new solutions per iteration, for a total of 100 function evaluations. We repeat each experiment with 5 independent random seeds, and Figure 3 shows the mean and standard deviation of the log-hypervolume difference (LHD) across the 20 post-initialization iterations. LHD is computed at each iteration as the logarithm of the difference between the maximum reachable hypervolume and the obtained hypervolume (see equation 40 in Appendix A.6). SPREAD delivers solid performance across the benchmark suite, achieving the lowest final values in most cases. It converges particularly rapidly on the 3-objective DTLZ2 and DTLZ5 problems and the 4-objective Car Side Impact problem. Notably, SPREAD consistently outperforms CDM-PSL, another diffusion-based generative method. This advantage stems from our novel conditioning strategy and our adaptive guidance mechanism, which steers samples more accurately toward the Pareto front, yielding stronger approximations overall. To assess the ability of both generative methods to fully solve MOBO problems, we compare their performance without employing SBX to escape local optima (step 8, Algorithm 3) in Appendix D (Figure 13), which demonstrates the superiority of our method.

## 6 CONCLUSION

We introduced SPREAD, a diffusion-based generative framework for multi-objective optimization that refines candidate solutions through adaptive, MGD-inspired guidance and a diversity-promoting repulsion mechanism. By integrating these components into a conditional diffusion process, SPREAD achieves both convergence toward Pareto optimality and broad coverage of the front. Experiments on

synthetic and real-world tasks show that SPREAD consistently outperforms state-of-the-art baselines in terms of hypervolume, diversity, and scalability, particularly in offline and Bayesian settings. A promising direction for future work is the design of a proper constraint-handling mechanism to extend SPREAD to multi-objective optimization problems with constraints on the decision variables.

## 7 ETHICS STATEMENT

This work does not involve human subjects, sensitive personal data, or applications with direct societal risks. The experiments are conducted entirely on publicly available benchmark problems and synthetic test functions commonly used in the multi-objective optimization community. No new datasets are collected, and all code and experimental protocols are designed for scientific reproducibility. We believe our contributions align with the ICLR Code of Ethics and do not raise ethical concerns beyond standard research integrity.

## 8 REPRODUCIBILITY STATEMENT

We have made significant efforts to ensure reproducibility of our results. All algorithmic details, hyper-parameter choices, and evaluation metrics are described in the main paper (Sections 4, 5) and the appendix (Appendices A–C). Complete proofs of theoretical results are provided in Appendix A. For all baseline methods, we rely on publicly available implementations or official repositories, as referenced in Appendix C. To facilitate full reproducibility, we release our implementation and scripts for reproducing all experimental results at: `https://github.com/safe-autonomous-systems/moo-spread`.

## 9 ACKNOWLEDGMENT

This project received funding from the German Federal Ministry of Education and Research (BMBF) through the AI junior research group "Multicriteria Machine Learning". All experiments were performed on the compute cluster of the Lamarr Institute for Machine Learning and Artificial Intelligence.

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

APPENDIX

## A ADDITIONAL PROOFS AND DETAILS

### A.1 PROOF OF THEOREM 1

We recall Theorem 1.

**Theorem 1.** (Objective Improvement) Let $\mathbf{X} \subset \mathcal{X}$ be a training dataset with distribution $P_{\mathbf{X}}$. Let $\Xi \in (0, \infty)^m$, independent of $\mathbf{X}$, and define the training label

$$\mathbf{C} := \mathbf{F}(\mathbf{X}) + \Xi. \tag{15}$$

For a conditioning value $\mathbf{c}$ in the support of $\mathbf{C}$, denote by $P_{\mathbf{X}|\mathbf{C}=\mathbf{c}}$ the true conditional data distribution and by $Q_\theta(\cdot \mid \mathbf{c})$ the distribution produced by a conditional DDPM when sampling conditioned on $\mathbf{c}$. Assume the sampler approximates the true conditional training distribution in total-variation TV distance by at most $\tau \in [0, 1)$:

$$\mathrm{TV}\big(Q_\theta(\cdot \mid \mathbf{c}), P_{\mathbf{X}|\mathbf{C}=\mathbf{c}}\big) = \sup_A |Q_\theta(A \mid \mathbf{c}) - P_{\mathbf{X}|\mathbf{C}=\mathbf{c}}(A)| \leq \tau. \tag{16}$$

Fix any initialization $\mathbf{x}_T \in \mathcal{X}$ and set $\mathbf{c} := \mathbf{c}_T = \mathbf{F}(\mathbf{x}_T)$. If $\mathbf{c}_T$ lies in the support of $\mathbf{C}$, and we draw $\mathbf{x}_0 \sim Q_\theta(\cdot \mid \mathbf{c}_T)$, then:

$$\mathbb{P}(\mathbf{x}_0 \prec \mathbf{x}_T) \geq 1 - \tau. \tag{17}$$

In other words, conditioning the reverse diffusion on $\mathbf{F}(\mathbf{x}_T)$ yields, with probability at least $1 - \tau$, a sample that dominates $\mathbf{x}_T$.

*Proof.* We proceed in two steps.

1. Characterization of the true conditional training distribution $P_{\mathbf{X}|\mathbf{C}=\mathbf{c}}$.

   Conditioning the training data on $\mathbf{C}$ forces $\mathbf{F}(\mathbf{X}) = \mathbf{C} - \Xi$. Because each component of $\Xi$ is strictly positive, we have $\mathbf{F}(\mathbf{X}) \prec \mathbf{C}$. Formally, letting $A_\mathbf{c} := \{\mathbf{x} \in \mathcal{X} : \mathbf{F}(\mathbf{x}) \prec \mathbf{c}\}$, we have

$$P_{\mathbf{X}|\mathbf{C}=\mathbf{c}}(A_\mathbf{c}) = 1. \tag{18}$$

   This means that, if we sample a point $\mathbf{x}$ from the training data distribution conditioned on a label $\mathbf{c}$ that lies in the support of $\mathbf{C}$, then the objective vector of that sample is almost surely strictly below $\mathbf{c}$ component-wise (because among the data points that carry the training label $\mathbf{c}$, essentially all of them have objective values strictly below $\mathbf{c}$).

2. Transfer guarantee from the true conditional training distribution to the learned sampler via TV.

The total-variation assumption (equation 16) says that for every measurable set $A$,

$$\big| Q_\theta(A \mid \mathbf{c}) - P_{\mathbf{X}\mid\mathbf{C}=\mathbf{c}}(A) \big| \leq \tau. \tag{19}$$

Simply put, when we condition on the label $\mathbf{c}$, the model's sampling probabilities are uniformly close to the true training data probabilities, within $\tau$, for every event we could ask about.

Applying equation 19 with $A = A_\mathbf{c}$ and using equation 18 yields

$$Q_\theta(A_\mathbf{c} \mid \mathbf{c}) \geq P_{\mathbf{X}\mid\mathbf{C}=\mathbf{c}}(A_\mathbf{c}) - \tau \tag{20}$$
$$Q_\theta(A_\mathbf{c} \mid \mathbf{c}) \geq 1 - \tau. \tag{21}$$

So, if we draw $\mathbf{x}_0 \sim Q_\theta(\cdot \mid \mathbf{c})$, then

$$\mathbb{P}(\mathbf{F}(\mathbf{x}_0) \prec \mathbf{c}) \geq 1 - \tau \tag{22}$$

Finally, assume we conditioned the sampler on $\mathbf{c} = \mathbf{F}(\mathbf{x}_T)$. Under the assumption "$\mathbf{c} = \mathbf{F}(\mathbf{x}_T)$ lies in the support of $\mathbf{C}$", all the conditional probabilities above are well-defined, and equation 22 immediately implies $\mathbb{P}(\mathbf{F}(\mathbf{x}_0) \prec \mathbf{F}(\mathbf{x}_T)) \geq 1 - \tau$. This completes the proof.

$\square$

To empirically support the dominance claim in Theorem 1, we provide a direct visualization of the initial points and their corresponding sampled points. Since dominance can be visually verified in two objectives, we conduct this experiment on the bi-objective problems considered in the online setting. Figure 4 presents the results for ZDT1, ZDT2, ZDT3, and RE21. No guidance mechanism is used so that the effect of the diffusion model alone can be clearly assessed. As shown in the plots, the sampled points consistently dominate their initializations, providing empirical validation of the theoretical improvement stated in Theorem 1.

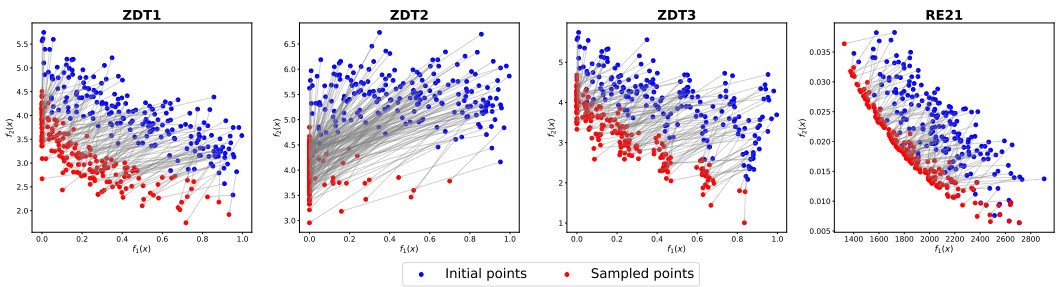

Figure 4: Empirical validation of the objective improvement guaranteed by Theorem 1. The figure shows the movement of random initial points (blue) toward the corresponding sampled points (red) on ZDT1, ZDT2, ZDT3 and RE21 in the online setting. *Guidance is disabled so that only the effect of the diffusion model is visualized.*

## A.2 PROOF OF THEOREM 2

We recall Theorem 2.

**Theorem 2.** Assume each objective function $f_j$ is continuously differentiable, and that $\nu_t = 0$ for all $t \in \{1, \ldots, T\}$. Let, at reverse timestep $t$,

$$a_{i,j} = \left\langle \nabla f_j(\mathbf{x}_t^{'i}), \mathbf{h}_t^i \right\rangle, \quad b_{i,j} = \left\langle \nabla f_j(\mathbf{x}_t^{'i}), \delta_t \right\rangle,$$

with $a_{i,j} > 0$ for all $i = 1, \dots, n$ and $j = 1, \dots, m$. Define

$$\gamma_t^i = \begin{cases} \rho \min\limits_{j:\, b_{i,j} < 0} \left( -\dfrac{a_{i,j}}{b_{i,j}} \right), \; 0 < \rho < 1, & \text{if any } b_{i,j} < 0, \\ \zeta, \quad \zeta > 0, & \text{otherwise,} \end{cases} \tag{23}$$

where $\rho$ controls the magnitude of the scaling parameters $\gamma_t^i$, and $\zeta$ denotes an arbitrary positive scalar. Then, $-\tilde{\mathbf{h}}_t^i = -\left( \mathbf{h}_t^i + \gamma_t^i \delta_t \right)$ serves as a common descent direction for all objectives at $\mathbf{x}_t^{'i}$.

*Proof.* For each point $\mathbf{x}_t^{'i}$ at reverse timestep $t$, define

$$a_{i,j} = \left\langle \nabla f_j(\mathbf{x}_t^{'i}), \mathbf{h}_t^i \right\rangle, \quad \text{and} \quad b_{i,j} = \left\langle \nabla f_j(\mathbf{x}_t^{'i}), \delta_t \right\rangle.$$

It suffices to prove that, for all $j = 1, \dots, m$:

$$\left\langle \nabla f_j(\mathbf{x}_t^{'i}), \tilde{\mathbf{h}}_t^i \right\rangle > 0 \tag{24}$$

$$\left\langle \nabla f_j(\mathbf{x}_t^{'i}), \left( \mathbf{h}_t^i + \gamma_t^i \delta_t \right) \right\rangle > 0 \tag{25}$$

$$a_{i,j} + \gamma_t^i b_{i,j} > 0. \tag{26}$$

Since $\nu_t = 0$, the main direction $\mathbf{h}_t^i$ is well aligned with the MGD direction at $\mathbf{x}_t^{'i}$, and thus inherits its descent property, i.e. $a_{i,j} > 0$ for all $j = 1, \dots, m$

- If $b_{i,j} > 0$ for all $j = 1, \dots, m$, then any choice of $\gamma_t^i = \zeta$ (with $\zeta > 0$) works.

- Otherwise, if $b_{i,j} < 0$ for some $j \in \{1, \dots, m\}$.

  For indices $j$ with $b_{i,j} < 0$, the inequality in (equation 26) is equivalent to

  $$\gamma_t^i < -\frac{a_{i,j}}{b_{i,j}}. \tag{27}$$

  For indices $j$ with $b_{i,j} > 0$, the inequality in (equation 26) is satisfied with any $\gamma_t^i > 0$.

  Therefore, a valid choice is

  $$0 < \gamma_t^i = \rho \min_{j,\, b_{i,j} < 0} \left( -\frac{a_{i,j}}{b_{i,j}} \right), \quad \text{with } 0 < \rho < 1, \tag{28}$$

  which ensures that the inequality in (equation 26) is satisfied for all $j = 1, \dots, m$.

$\square$

## A.3 A DISCUSSION ON $\nu_t > 0$ IN THEOREM 2

As outlined in the main text, the assumption that we turn of the diversity criterion for the Pareto set by choosing $\nu_t = 0$ in Theorem 2 is quite restrictive and limiting. It is possible to prove an alternative version of Theorem 2 when dropping this assumption. Since this would lead to the requirement of a sample-wise online adaptation of the penalty parameter $\nu_t$ as well as significant additional computational cost, we have decided to pursue this approach in practice. However, a proof would follow along the arguments laid out next.

First, we note that in the optimization problem in equation 13, the first term $-1/n \sum_{i=1}^n \langle \mathbf{g}_t^i, \mathbf{u}^i \rangle$ can simply be decomposed into a sample-wise formulation, where we try to align the descent direction $\mathbf{u}^i$ as much with the multi-objective steepest descent direction $\mathbf{g}_t^i$. The coupling occurs only in the second term, where we try to diversify the directions using the repulsion term $\Gamma_t$ from equation 12. Setting $\gamma_t = 0$ thus simply leads to $\mathbf{u}^i = \mathbf{g}_t^i$ for all samples $i$. Setting $\nu_t > 0$ thus leads to a divergence between the two, and we need to bound the maximum angle between $\mathbf{u}^i$ and $\mathbf{g}_t^i(\mathbf{x})$ in order to ensure that $\mathbf{u}^i$ remains a common descent direction for sample $i$. For simplicity, we are now going to drop the indices $i$ and $t$ in the following.

Next, following the arguments above, we need to ensure that the angle between any individual gradient $\nabla f_j(\mathbf{x})$ and $\mathbf{u}$ remains sufficiently large. In Peitz & Dellnitz (2018a), an additional a-posteriori criterion was derived for the optimization problem (4), in which the weights $\lambda_j^*$ are derived that ultimately lead to the construction $\mathbf{g}(\mathbf{x}) = \sum_{j=1}^m \lambda_j^* \nabla f_j(\mathbf{x})$. To bound the angle between descent direction $\mathbf{g}(\mathbf{x})$ and $\nabla f_j(\mathbf{x})$ from above (i.e., to make it strictly smaller than the otherwise allowed maximum $\pi/2$ for a non-increasing direction), the corresponding weight $\lambda_j^*$ has to be larger than a lower bound $\lambda_{j,\min}^*$ that can be computed when knowing $\mathbf{g}(\mathbf{x})$ as well as all the $\nabla f_j(\mathbf{x})$. As a consequence, one obtains a bound of the form $\|\mathbf{g}(\mathbf{x}) - \nabla f_j(\mathbf{x})\|_2 < \epsilon_j$.

Our strategy is now to bound the second term in equation 13 from above such that the difference between $\mathbf{g}(\mathbf{x})$ and $\mathbf{u}$ is smaller than the smallest difference between $\mathbf{g}(\mathbf{x})$ and $\nabla f_j(\mathbf{x})$,

$$\|\mathbf{g}(\mathbf{x}) - \mathbf{u}\| \le \max_{j \in \{1,\ldots,m\}} \|\mathbf{g}(\mathbf{x}) - \nabla f_j(\mathbf{x})\|_2 < \epsilon_j.$$

We observe that the maximum value for $\Gamma_t$ in equation 12 is one in the case where all samples coincide. We thus have $\Gamma_t \in [0,1]$, and consequently $(\nu_t \Gamma_t) \in [0, \nu_t]$. Since the optimization problem (13) trades between the deviation of $\mathbf{u}$ from $\mathbf{g}(\mathbf{x})$ and the spreading, we find that the inner product $\langle \mathbf{g}(\mathbf{x}), \mathbf{u} \rangle$ (and thus the angle) is also bounded.

In summary, an appropriate choice of $\nu_t$ leads to a maximum deviation between $\mathbf{u}$ and $\mathbf{g}(\mathbf{x})$. By making sure that this deviation is smaller than the minimal angle between $\mathbf{g}(\mathbf{x})$ and $\nabla f_j(\mathbf{x})$, descent can be guaranteed. However, in practice this would require us to perform a costly calculation and sample-wise adaptation of $\nu_t$, which proves to be impractical. Moreover, we find that allowing a temporary increase in favor of a better diversity is in the end beneficial for the Pareto set coverage.

### A.4 EXTENDED ARCHITECTURAL DETAILS

Figure 1 shows our noise-prediction network DiT-MOO, conditioned on the objective values via a multi-head cross-attention module (MH Cross-Attention: `MHCA`). The input projection, time embedding, and condition embedding are implemented as linear layers with hidden dimension $e$, while the output projection is a linear layer with input dimension $e$. At timestep $t$, let $\mathbf{Z}_t \in \mathbb{R}^{n \times 1 \times e}$ (layer-normalized features from $\mathbf{X}_t$) and $\mathbf{B}_t \in \mathbb{R}^{n \times 2 \times e}$ (the concatenation of the condition and time embeddings) be the inputs to `MHCA`. With $h$ attention heads, where each head has dimension $d_k = d_v = e/h$, the `MHCA` module computes, for each head $i \in \{1, \ldots, h\}$:

$$\mathbf{Q}^{(i)} = \mathbf{Z}_t \mathbf{W}_Q^{(i)} \in \mathbb{R}^{n \times 1 \times d_k},$$
$$\mathbf{K}^{(i)} = \mathbf{B}_t \mathbf{W}_K^{(i)} \in \mathbb{R}^{n \times 2 \times d_k},$$
$$\mathbf{V}^{(i)} = \mathbf{B}_t \mathbf{W}_V^{(i)} \in \mathbb{R}^{n \times 2 \times d_v},$$

where $\mathbf{W}_Q^{(i)} \in \mathbb{R}^{e \times d_k}$, $\mathbf{W}_K^{(i)} \in \mathbb{R}^{e \times d_k}$, and $\mathbf{W}_V^{(i)} \in \mathbb{R}^{e \times d_v}$ are learnable projections[1].

The attention weights are

$$\mathbf{A}^{(i)} = \mathrm{softmax}\left( \frac{\mathbf{Q}^{(i)}(\mathbf{K}^{(i)})^\top}{\sqrt{d_k}} \right) \in \mathbb{R}^{n \times 1 \times 2}, \tag{29}$$

and the head output is

$$\mathbf{O}^{(i)} = \mathbf{A}^{(i)} \mathbf{V}^{(i)} \in \mathbb{R}^{n \times 1 \times d_v}. \tag{30}$$

Finally, the head outputs are concatenated and projected:

$$\mathrm{MHCA}(\mathbf{Z}_t, \mathbf{B}_t) = \left( \mathrm{concat}_{i=1}^h \mathbf{O}^{(i)} \right) \mathbf{W}_O \in \mathbb{R}^{n \times 1 \times e}, \tag{31}$$

where $\mathbf{W}_O \in \mathbb{R}^{e \times e}$ is a learnable projection matrix.

For a fixed number of blocks $L$, the number of parameters in DiT-MOO depends on the input dimension $d$ and the objective space dimension $m$ of the considered multi-objective optimization problem. In all our experiments with SPREAD, we set $L = 3$, which yields a total of approximately 800k parameters (ranging from $797{,}571$ to $804{,}894$).

---

[1] Here, we use the convention that a tensor in $\mathbb{R}^{n \times a \times b}$ represents $n$ matrices of size $a \times b$, and matrix multiplications are carried out in parallel across the batch dimension.

A.5   ADDITIONAL METHODOLOGICAL DETAILS

**Latin Hypercube Sampling (LHS)**   Latin hypercube sampling is a stratified sampling technique for generating well-distributed initial points in $\mathbb{R}^d$ (McKay et al., 2000). Given a sample size $N$, the range of each decision variable $x_j$ ($j = 1, \ldots, d$) is partitioned into $N$ disjoint intervals of equal probability under the uniform distribution. One value is drawn uniformly at random from each interval, yielding $N$ candidate values per dimension. The values across dimensions are then randomly permuted and paired, so that each sample $\mathbf{x}^i = (x_1^i, \ldots, x_d^i) \in \mathcal{X} \subset \mathbb{R}^d$ contains exactly one value from each interval of every variable. This makes LHS particularly useful for covering the decision space uniformly with relatively few samples.

**Cosine Variance Schedule**   The cosine variance schedule (Nichol & Dhariwal, 2021) is a technique for defining the forward diffusion noise schedule in a smooth, non-linear fashion. Instead of linearly increasing the variance $\beta_t$ over timesteps $t = 1, \ldots, T$, the cumulative product of the noise-retention coefficients, $\bar{\alpha}_t = \prod_{s=1}^{t}(1 - \beta_s)$, is parameterized using a shifted cosine function:

$$\bar{\alpha}_t = \frac{\cos^2\left(\frac{t/T+s}{1+s} \cdot \frac{\pi}{2}\right)}{\cos^2\left(\frac{s}{1+s} \cdot \frac{\pi}{2}\right)}, \quad t = 0, \ldots, T, \tag{32}$$

where $s \geq 0$ is a small offset to avoid singularities near $t = 0$. The corresponding variances $\beta_t$ are then recovered from $\bar{\alpha}_t$. Compared to linear schedules, the cosine schedule allocates more steps to low-noise regions, resulting in improved sample quality and training stability in practice.

**Armijo Backtracking Line Search**   At each timestep $t$ of the sampling process in SPREAD, the step size $\eta_t$ is determined using the Armijo backtracking line search (Armijo, 1966). Given a search direction $\tilde{\mathbf{h}}_t(\mathbf{X}'_t)$, we start from an initial step size $\eta = \eta_0$ and iteratively reduce it by a factor $b \in (0, 1)$ until the Armijo condition is satisfied:

$$\mathbf{F}(\mathbf{X}'_t - \eta\tilde{\mathbf{h}}_t(\mathbf{X}'_t)) \leq \mathbf{F}(\mathbf{X}'_t) - a\eta\nabla\mathbf{F}(\mathbf{X}'_t)^\top\tilde{\mathbf{h}}_t(\mathbf{X}'_t), \tag{33}$$

where $a \in (0, 1)$ is a fixed parameter. This condition ensures a sufficient decrease in the objective function while avoiding overly aggressive steps. We set $a = 10^{-4}$ and $b = 0.9$ in our experiments.

**Crowding Distance**   The crowding distance (Deb et al., 2002a) is a density estimator widely used in evolutionary multi-objective optimization to preserve diversity along the Pareto front. Given a non-dominated set $\mathcal{S} = \{\mathbf{x}^1, \ldots, \mathbf{x}^n\}$, the crowding distance of solution $\mathbf{x}^i \in \mathbb{R}^d$ is computed by summing the normalized objective-wise distances to its immediate neighbors. For each objective $j \in \{1, \ldots, m\}$, sort the solutions by $f_j$, and assign infinite distance to the boundary solutions. For interior points, the contribution in objective $j$ is

$$d_j^i = \frac{f_j(\mathbf{x}^{i+1}) - f_j(\mathbf{x}^{i-1})}{\max_k f_j(\mathbf{x}^k) - \min_k f_j(\mathbf{x}^k)}. \tag{34}$$

The overall crowding distance of $\mathbf{x}^i$ is then

$$\mathrm{CD}(\mathbf{x}^i) = \sum_{j=1}^{m} d_j^i, \tag{35}$$

with $\mathrm{CD}(\mathbf{x}^i) = +\infty$ for boundary points. Larger crowding distances indicate that a solution lies in a less crowded region of the objective space, making it more likely to be selected.

**Computational and Memory Complexity Analysis**   Let $K$ be the number of gradient steps used to solve the sub-problem in equation 13. In each reverse step of Algorithm 1, SPREAD performs four main operations: performing a DiT-MOO denoising pass, computing the multi-gradient descent directions, solving the diversity-regularized sub-problem, and applying the guidance update. The denoising update uses a single forward pass through DiT-MOO, which processes each sample independently and costs $\mathcal{O}(n(d + m))$. Computing the MGD directions requires evaluating all objective gradients $\nabla f_j(\mathbf{x}_t^i)$, yielding a cost of $\mathcal{O}(nmd)$. Solving the sub-problem in equation 13 for

$K$ gradient steps requires evaluating the objectives for all points, $\mathcal{O}(nmd)$, together with computing the pairwise RBF-based repulsion term in objective space, $\mathcal{O}(n^2 m)$. Thus the full sub-problem costs $\mathcal{O}(K(nmd + n^2 m))$. Finally, the guidance update involves only vector projections and costs $\mathcal{O}(nd)$. Summing these contributions and keeping only the dominant terms, one reverse-diffusion step has complexity

$$\mathcal{O}\big(nmd + K(nmd + n^2 m)\big), \tag{36}$$

and running all $T$ steps yields the total sampling complexity

$$\mathcal{O}\left(Tnmd + TK(nmd + n^2 m)\right), \tag{37}$$

which is dominated by the $\mathcal{O}(TKn^2 m)$ term when objective evaluations are inexpensive and $n$ is large.

The memory usage during sampling is determined by storing the $n$ points ($\mathcal{O}(nd)$), their objective values ($\mathcal{O}(nm)$), the pairwise distances required for the repulsion term ($\mathcal{O}(n^2 m)$), and the DiT-MOO activations ($\mathcal{O}(neL)$), in addition to the model parameters. Therefore, the total memory scales as

$$\mathcal{O}\big(n(d + m) + n^2 m + neL + |\theta|\big). \tag{38}$$

## A.6    EVALUATION METRICS

**Hypervolume (HV)**    The hypervolume indicator measures the portion of objective space that is weakly dominated by the approximated Pareto front with respect to a fixed reference point (Zitzler & Thiele, 2002). Formally, for a set of non-dominated solutions $\mathcal{P} = \{\mathbf{x}^i\}_{i=i}^{n}$ with $\mathbf{x}^i \in \mathbb{R}^d$, the hypervolume is defined as

$$\mathrm{HV}(\mathcal{P}) = \Lambda\left(\left\{\mathbf{a} \in \mathbb{R}^d \mid \exists\, \mathbf{x} \in \mathcal{P} \,:\, \mathbf{a} \in \prod_{j=1}^{m}[f_j(\mathbf{x}), \mathbf{r}]\right\}\right) \tag{39}$$

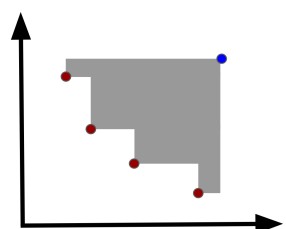

where $\mathbf{r} \in \mathbb{R}^m$ is a reference point dominated by all Pareto optimal solutions, and $\Lambda(\cdot)$ is the Lebesgue measure (see Figure 5). We report in Appendix C, the reference points used in our experiments. The HV is maximized when the solution set covers the Pareto front broadly and accurately, making it a widely used indicator for comparing multi-objective optimization methods.

Figure 5: **Hypervolume** bi-objective example, corresponding to the shaded region defined by the obtained solutions (red) and the reference point (blue).

**Log Hypervolume Difference (LHD)**    The log hypervolume difference is a commonly used indicator to assess the convergence of multi-objective Bayesian optimization methods. Let $\mathrm{HV}^*$ denote the maximum reachable hypervolume (i.e., the hypervolume of the true Pareto front). At iteration $t$, let $\mathcal{P}_t$ be the approximated Pareto front obtained by the algorithm. The LHD is then defined as:

$$\mathrm{LHD}_t = \log(\mathrm{HV}^* - \mathrm{HV}(\mathcal{P}_t)). \tag{40}$$

A lower LHD value indicates that the current solution set is closer to the optimal hypervolume.

**$\Delta$-spread**    The $\Delta$-spread (Deb et al., 2002a) evaluates the diversity of an approximated Pareto front by comparing the spacing between consecutive solutions to the average spacing, while also accounting for the coverage of the true extreme points. Let $\mathcal{Y} = \{\mathbf{F}(\mathbf{x}^1), \ldots, \mathbf{F}(\mathbf{x}^n)\}$ denote the non-dominated solutions, sorted along a chosen objective. Define $d_i = \|\mathbf{F}(\mathbf{x}^{i+1}) - \mathbf{F}(\mathbf{x}^i)\|$ as the Euclidean distance between consecutive solutions, $\bar{d} = \frac{1}{n-1}\sum_{i=1}^{n-1} d_i$ as their mean, and $d_f, d_l$ as the distances from the extreme solutions in $\mathcal{Y}$ to the true Pareto front endpoints (if available, otherwise set to 0). The $\Delta$-spread is given by:

$$\Delta\text{-spread} = \frac{d_f + d_l + \sum_{i=1}^{n-1} |d_i - \bar{d}|}{d_f + d_l + (n-1)\bar{d}}. \tag{41}$$

A lower value indicates a more uniform spread of solutions along the Pareto front. By convention, $\Delta$-spread $= +\infty$ if the solution set collapses to a single point.

## B SPREAD IN THE MOBO SETTING

In a MOBO framework, the true objectives are expensive to evaluate, so surrogate models (e.g., Gaussian processes) are trained on an initial dataset of evaluated solutions. At each iteration, the surrogate models are updated with newly evaluated solutions, and a search strategy proposes new candidate solutions to evaluate (Paria et al., 2020; Lin et al., 2022). This iterative cycle of modeling, proposing, and evaluating continues until the evaluation budget is exhausted, yielding an approximation of the Pareto front. To adapt our method to this setting, we use SPREAD as the search strategy for proposing new candidate solutions. The full procedure is given in Algorithm 3. Following (Li et al., 2025a), we employ simulated binary crossover (SBX) as an auxiliary operator to escape local minima, i.e., when no improvement is observed for a fixed number of iterations. Batch selection is then performed using the hypervolume metric, as in (Lin et al., 2022). To increase the number of training samples per iteration $k$, we adopt the data augmentation strategy of (Li et al., 2025a). Specifically, data points are first extracted from $\mathbf{X}^{(k)}$ using shift-based density estimation (Li et al., 2013). Three transformations are then applied: small random perturbations, interpolation of randomly chosen pairs, and Gaussian noise injection. The augmented samples are shuffled, truncated to match the target augmentation factor, and merged with the extracted points to form the enhanced dataset used for training DiT-MOO.

**Simulated Binary Crossover (SBX)**  Simulated Binary Crossover (SBX) (Deb, 1995) is a real-parameter recombination operator used to generate two offspring from two parents. Given parent vectors $p_1, p_2 \in \mathbf{X}^{(k)} \subset \mathbb{R}^d$ and a distribution index parameter $\varkappa > 0$, SBX first samples a random vector $u \in [0, 1]^d$, then computes a spread factor $\tau_j$ for each $j = 1, \ldots, d$ as

$$\tau_j = \begin{cases} (2u_j)^{\frac{1}{\varkappa+1}}, & u_j \leq 0.5, \\ \left(\frac{1}{2(1-u_j)}\right)^{\frac{1}{\varkappa+1}}, & u_j > 0.5. \end{cases} \tag{42}$$

The two offspring are then set as

$$\begin{aligned} \text{offspring1}_j &= 0.5\big((1+\tau_j)\,p_{1,j} + (1-\tau_j)\,p_{2,j}\big), \\ \text{offspring2}_j &= 0.5\big((1-\tau_j)\,p_{1,j} + (1+\tau_j)\,p_{2,j}\big). \end{aligned} \tag{43}$$

A higher $\varkappa$ makes offspring closer to the parents (less exploratory), while lower $\varkappa$ allows more distant (diverse) offspring. This operation is repeated 1000 times, and the resulting points are passed to the batch selection step (Step 10 in Algorithm 3). In our experiments, we use $\varkappa = 15$.

---

**Algorithm 3** MOBO with SPREAD

---

**Input:** DiT-MOO as the noise prediction network $\hat{\epsilon}_\theta(\cdot)$, a black-box multi-objective function $\mathbf{F}(\cdot)$ defined on $\mathcal{X}$.
**Parameter:** initial sample size $n_{\text{init}}$, number of iterations $K$, batch size $b$, a boolean flag `escape` (initialized to `False`).
**Output:** approximate Pareto optimal points.

1: Get the initial solutions $\{\mathbf{x}^{(0)i}\}_{i=1}^{n_{\text{init}}} = \mathbf{X}^{(0)}$ via LHS, and evaluate $\mathbf{Y}^{(0)} = \mathbf{F}(\mathbf{X}^{(0)})$.
2: **for** $k = 0$ **to** $K - 1$ **do**
3:     Train Gaussian-Process surrogates $\{\text{gp}_j^k\}_{j=1}^m$ using $\{\mathbf{X}^{(k)}, \mathbf{Y}^{(k)}\}$.
4:     $\mathbf{X}_{\text{train}}^{(k)} \leftarrow$ Get training data for DiT-MOO using $\{\mathbf{X}^{(k)}, \mathbf{Y}^{(k)}\}$ as in CDM-PSL (Li et al., 2025a) (Appendix B)
5:     **if** `escape` is `False` **then**
6:         $\boldsymbol{S} \leftarrow$ Generate offspring with SPREAD via Algorithm 4).
7:     **else**
8:         $\boldsymbol{S} \leftarrow$ Apply SBX to escape local-optima (Appendix B).
9:     **end if**
10:    $\mathbf{X}_{\text{new}}^{(k)} \leftarrow$ Batch selection: select the top-$b$ solutions from $\boldsymbol{S}$ based on their hypervolume contributions.
11:    $\mathbf{Y}^{(k+1)} \leftarrow \mathbf{Y}^{(k)} \cup \mathbf{F}(\mathbf{X}_{\text{new}}^{(k)})$
12:    $\mathbf{X}^{(k+1)} \leftarrow \mathbf{X}^{(k)} \cup \mathbf{X}_{\text{new}}^{(k)}$
13:    Decide whether to invert `escape` based on the latest hypervolume values.
14: **end for**
**Return:** $\mathbf{X}^K$

---

---

**Algorithm 4** Offspring generation with SPREAD (MOBO setting)

---

**Input:** DiT-MOO as the noise prediction network $\hat{\epsilon}_\theta(\cdot)$, a black-box multi-objective function $\mathbf{F}(\cdot)$ defined on $\mathcal{X}$, a training dataset $\mathbf{X}_{\text{train}}^{(k)}$, Gaussian-Process surrogates $(\text{gp}_j^k)_{j=1}^m = \text{GP}$.
**Parameter:** epochs $E$, timesteps $T$, number of generation $N_{\text{gen}}$, required offspring size $n$.
**Output:** offspring $S$ generated by SPREAD.

1: Train $\hat{\epsilon}_\theta(\cdot)$ for $E$ epochs on $\mathbf{X}_{\text{train}}^{(k)}$ via Algorithm 2 using GP instead of $\mathbf{F}$.
2: $S \leftarrow \emptyset$
3: **for** $i = 1$ to $N_{\text{gen}}$ **do**
4:     Initialize $n$ random points $\mathbf{X}_T = \{\mathbf{x}_T^i\}_{i=1}^n \subset \mathcal{X}$
5:     $\mathcal{P}_T \leftarrow \mathbf{X}_T$
6:     **for** $t = T$ **to** 1 **do**
7:         $(\mathbf{g}_t^i)_{i=1}^n \leftarrow$ Get the MGD directions via Section 3.3 using GP instead of $\mathbf{F}$.
8:         $(\mathbf{h}_t^i)_{i=1}^n \leftarrow$ Get the main directions via equation 13 using GP instead of $\mathbf{F}$.
9:         $(\tilde{\mathbf{h}}_t^i)_{i=1}^n \leftarrow$ Get the guidance directions via equation 10.
10:       $\mathbf{X}_{t-1} \longleftarrow$ Get the denoised points via equation 9.
11:       $\mathcal{P}_{t-1} \leftarrow$ Use crowding distance (Appendix A.5) to get the top-$n$ non-dominated points
          from $\mathbf{X}_{t-1} \cup \mathcal{P}_t$.
12:     **end for**
13:     $S \leftarrow S \cup \{\mathcal{P}_0\}$
14: **end for**
**Return:** $S$

---

## C   Implementation Details

In this section, we provide further details about the experimental settings. For each training batch $\mathbf{X} = \{\mathbf{x}^i\}_{i=1}^{N_b}$, we set $\Xi = \omega \cdot \mathbf{1}_m$, where $\mathbf{1}_m$ denotes the $m$-dimensional vector of ones. The value of $\omega$ is determined as

$$\omega = \begin{cases} \min\{\mathbf{F}(\mathbf{x}^i) \mid \mathbf{x}^i \in \mathbf{X}, \ \mathbf{F}(\mathbf{x}^i) > 0\}, & \text{if this set is nonempty,} \\ 10^{-6}, & \text{otherwise.} \end{cases}$$

In all experiments, we fix the number of DiT blocks to $L = 3$. DiT-MOO is trained for a maximum of 1000 epochs with early stopping after 100 epochs, except in the Bayesian MOO setting, where a maximum of 250 epochs is used. The number of solutions produced by all methods is 200 in the main experiments and 256 in the offline setting. For the Bayesian MOO setting, 5 solutions are selected at each of the 20 steps. We consider $T = 5000$ timesteps for the main experiments, and 1000 and 25 timesteps for the offline and Bayesian settings, respectively. At each timestep, we solve for the main directions $(\mathbf{h}_t^i)_{i=1}^n$ using gradient descent with 10 iterations and a fixed $\nu_t = 10$. In the repulsion function, the length scale $\sigma$ is set adaptively from the pairwise squared distances as

$$2\sigma^2 = 5 \cdot 10^{-6} \times \frac{\text{median}\big(\{\|\mathbf{F}_i - \mathbf{F}_j\|^2 : 1 \le i, j \le n\}\big)}{\log n}, \tag{44}$$

where $\mathbf{F}_i \in \mathbb{R}^m$ are the objective vectors in the batch of samples and $n$ is the number of samples.[2] In all experiments, we use fixed values of $\rho$ determined solely by the number of objectives and, in a few cases, by the specific problem instance:

- Online setting:
    - $m = 2$ : $\rho = 0.9$
    - $m > 2$ : $\rho = 0.001$
- Offline setting:
    - $m = 2$ : $\rho = 0.9$, except for ZDT4($\rho = 0.0001$), and ZDT6($\rho = 0.1$).
    - $m > 2$: $\rho = 0.001$, except for DTLZ4, and DTLZ6: $\rho = 0.0001$.
- Bayesian setting:
    - $m = 2$ : $\rho = 0.9$

---

[2]In implementation, we follow a PyTorch workaround from Liu et al. (2021) to simulate NumPy's `median`.

    – $m > 2 : \rho = 0.01$

We determined the default $\rho$ values from a few preliminary checks and used them unchanged throughout our experiments. We did not conduct any extensive hyperparameter search. While Figure 10 shows that SPREAD's performance varies across a wide range of $\rho$ values, the relative ordering with respect to the baselines remains stable: even with non-optimal choices of $\rho$, SPREAD does not exhibit any drastic performance degradation relative to the competing methods. For new problems, we recommend using the default values, which depend only on the number of objectives. In the main experiments, we use the implementations provided by the PyTorch library LibMOON (Zhang et al., 2024a)[3] for HVGrad, PMGDA, and STH. For MOO-SVGD, we rely on the authors' official code.[4] In the offline setting, implementation and evaluation protocols follow Off-MOO-Bench (Xue et al., 2024), and we adopt baseline results from Annadani et al. (2025). Five independent seeds $(1000, 2000, \ldots, 5000)$ are used, and we report mean and standard deviation across runs. The same seeds are used across all experiments in all settings, and for ablation studies where mean and standard deviation are not reported, we fix the seed to $1000$. In the Bayesian MOO setting, we use the publicly available codes provided by the respective authors. We report in Tables 5, 6, 7 and 8 detailed information about the synthetic and real-world problems considered across the different settings. The experiments were run on a single NVIDIA A100-SXM4-40GB GPU. To facilitate reproducibility, our code is available at the following anonymous repository: `https://anonymous.4open.science/r/SPREAD-2E32`.

Table 5: **Benchmark problems.** Problem settings and reference points in the main experiments.

| Name | $d$ | $m$ | Type | Pareto Front Shape | Reference Point for Hypervolume Computation |
|---|---|---|---|---|---|
| ZDT1 | 30 | 2 | Continuous | Convex | (0.9994, 6.0576) |
| ZDT2 | 30 | 2 | Continuous | Concave | (0.9994, 6.8960) |
| ZDT3 | 30 | 2 | Continuous | Disconnected | (0.9994, 6.0571) |
| DTLZ2 | 30 | 3 | Continuous | Concave | (2.8390, 2.9011, 2.8575) |
| DTLZ4 | 30 | 3 | Continuous | Concave | (3.2675, 2.6443, 2.4263) |
| DTLZ7 | 30 | 3 | Continuous | Disconnected | (0.9984, 0.9961, 22.8114) |
| RE21 (Four bar truss design) | 4 | 2 | Continuous | Convex | (3144.44, 0.05) |
| RE33 (Disc brake design) | 4 | 3 | Continuous | Unknown | (5.01, 9.84, 4.30) |
| RE34 (Vehicle crashworthiness design) | 5 | 3 | Continuous | Unknown | (1.86472022e+03, 1.18199394e+01, 2.90399938e-01) |
| RE37 (Rocket injector design) | 4 | 3 | Continuous | Unknown | (1.1022, 1.20726899, 1.20318656) |
| RE41 (Car side impact design) | 7 | 4 | Continuous | Unknown | (47.04480682, 4.86997366, 14.40049127, 10.3941957) |

Table 6: **Offline MOO benchmarks:** task properties.

| Task Name | Dataset size | Dimensions | # Objectives | Search space |
|---|---|---|---|---|
| Synthetic Function | 60000 | 7-30 | 2-3 | Continuous |
| Real-world Application | 60000 | 3-7 | 2-6 | Continuous, Integer & Mixed |

## D   ADDITIONAL RESULTS

**MGD+RBF Baseline and Guidance Ablation Experiments** We present experiments using the straightforward MGD+RBF baseline, as well as an ablation of SPREAD without any guidance mechanism. For MGD+RBF, we apply the second update of equation 9 for $T$ iterations (using the same $T$ as in the main experiments) to refine random initializations, without using any diffusion model. For the SPREAD ablation without guidance, we discard the second update of equation 9 and rely solely on the reverse-diffusion update (the first update of equation 9). As shown in Table 9, SPREAD consistently dominates both baselines in the online setting. In the offline setting (Table 10), SPREAD(w/o guidance) achieves the best performance on many DTLZ problems, while SPREAD surpasses both baselines on most ZDT and RE problems; MGD+RBF, however, shows strong performance on the RE61 task. In the MOBO setting, Figure 6 demonstrates that SPREAD and SPREAD(w/o guidance) both outperform MGD+RBF on most tasks, with SPREAD performing best overall. These results indicate that both diffusion and guidance are essential components contributing to the strong performance of SPREAD.

---

[3]`https://github.com/xzhang2523/libmoon`
[4]`https://github.com/gnobitab/MultiObjectiveSampling`

Table 7: **Offline MOO benchmarks:** problem settings and reference points.

| Name | $d$ | $m$ | Type | Pareto Front Shape | Reference Point for Hypervolume Computation |
|---|---|---|---|---|---|
| ZDT1 | 30 | 2 | Continuous | Convex | (1.10, 8.58) |
| ZDT2 | 30 | 2 | Continuous | Concave | (1.10, 9.59) |
| ZDT3 | 30 | 2 | Continuous | Disconnected | (1.10, 8.74) |
| ZDT4 | 10 | 2 | Continuous | Convex | (1.10, 300.42) |
| ZDT6 | 10 | 2 | Continuous | Concave | (1.07, 10.27) |
| DTLZ1 | 7 | 3 | Continuous | Linear | (558.21, 552.30, 568.36) |
| DTLZ2 | 10 | 3 | Continuous | Concave | (2.77, 2.78, 2.93) |
| DTLZ3 | 10 | 3 | Continuous | Concave | (1703.72, 1605.54, 1670.48) |
| DTLZ4 | 10 | 3 | Continuous | Concave | (3.03, 2.83, 2.78) |
| DTLZ5 | 10 | 3 | Continuous | Concave (2d) | (2.65, 2.61, 2.70) |
| DTLZ6 | 10 | 3 | Continuous | Concave (2d) | (9.80, 9.78, 9.78) |
| DTLZ7 | 10 | 3 | Continuous | Disconnected | (1.10, 1.10, 33.43) |
| RE21 (Four bar truss design) | 4 | 2 | Continuous | Convex | (3144.44, 0.05) |
| RE22 (Reinforced concrete beam design) | 3 | 2 | Mixed | Mixed | (829.08, 2407217.25) |
| RE25 (Coil compression spring design) | 3 | 2 | Mixed | Mixed, Disconnected | (124.79, 10038735.00) |
| RE31 (Two bar truss design) | 3 | 3 | Continuous | Unknown | (808.85, 6893375.82, 6793450.00) |
| RE32 (Welded beam design) | 4 | 3 | Continuous | Unknown | (290.66, 16552.46, 388265024.00) |
| RE33 (Disc brake design) | 4 | 3 | Continuous | Unknown | (8.01, 8.84, 2343.30) |
| RE35 (Speed reducer design) | 7 | 3 | Mixed | Unknown | (7050.79, 1696.67, 397.83) |
| RE36 (Gear train design) | 4 | 3 | Integer | Concave, Disconnected | (10.21, 60.00 , 0.97) |
| RE37 (Rocket injector design) | 4 | 3 | Continuous | Unknown | (0.99, 0.96, 0.99) |
| RE41 (Car side impact design) | 7 | 4 | Continuous | Unknown | (42.65, 4.43, 13.08, 13.45) |
| RE42 (Conceptual marine design) | 6 | 4 | Continuous | Unknown | (-26.39, 19904.90, 28546.79, 14.98) |
| RE61 (Water resource planning) | 3 | 6 | Continuous | Unknown | (83060.03, 1350.00, 2853469.06,16027067.60, 357719.74, 99660.36) |

Table 8: **Bayesian MOO benchmarks:** problem settings and reference points.

| Name | $d$ | $m$ | Type | Pareto Front Shape | Reference Point for Hypervolume Computation |
|---|---|---|---|---|---|
| ZDT1 | 20 | 2 | Continuous | Convex | (0.9994, 6.0576) |
| ZDT2 | 20 | 2 | Continuous | Concave | (0.9994, 6.8960) |
| ZDT3 | 20 | 2 | Continuous | Disconnected | (0.9994, 6.0571) |
| DTLZ2 | 20 | 3 | Continuous | Concave | (2.8390, 2.9011, 2.8575) |
| DTLZ5 | 20 | 3 | Continuous | Concave | (2.6672, 2.8009, 2.8575) |
| DTLZ7 | 20 | 3 | Continuous | Disconnected | (0.9984, 0.9961, 22.8114) |
| Branin and Currin | 2 | 2 | Continuous | Convex | (18.0, 6.0) |
| Penicillin Production | 7 | 3 | Continuous | Unknown | (1.8500, 86.9300, 514.7000) |
| Car Side Impact (RE41) | 7 | 4 | Continuous | Unknown | (45.4872, 4.5114, 13.3394, 10.3942) |

Table 9: (Online) MGD+RBF baseline and ablation study without guidance. The random seed is fixed to 1000, and the best results are highlighted in bold.

| Problem | SPREAD | | MGD+RBF | | SPREAD (w/o guidance) | |
|---|---|---|---|---|---|---|
| | HV | $\Delta$-spread | HV | $\Delta$-spread | HV | $\Delta$-spread |
| ZDT1 | **5.72** | **0.32** | 5.06 | 1.00 | 4.08 | 0.84 |
| ZDT2 | **6.22** | **0.30** | 5.89 | 1.00 | 4.00 | 0.85 |
| ZDT3 | **6.10** | **0.53** | 5.06 | 1.00 | 4.18 | 0.86 |
| RE21 | **70.11** | **0.41** | 32.84 | 1.00 | 70.05 | 0.45 |
| DTLZ2 | **22.92** | **0.92** | 0.00 | 1.63 | 22.25 | 0.94 |
| DTLZ4 | **20.22** | **0.76** | 18.84 | 1.14 | 17.88 | 1.37 |
| DTLZ7 | **18.08** | **0.61** | 16.72 | 1.00 | 11.88 | 0.74 |
| RE33 | **135.26** | **0.97** | 119.22 | 1.11 | 0.00 | 1.00 |
| RE34 | **242.69** | 0.93 | 210.07 | 1.14 | 237.07 | **0.78** |
| RE37 | **1.42** | **0.81** | 0.23 | 1.70 | 1.37 | 0.87 |
| RE41 | **1005.08** | **0.88** | 382.80 | 1.28 | 860.07 | 0.94 |

Table 10: (Offline) MGD+RBF baseline and ablation study without guidance. The random seed is fixed to 1000, and the best results are highlighted in bold.

| Method | ZDT1 | ZDT2 | ZDT3 | ZDT4 | ZDT6 | DTLZ1 | DTLZ2 | DTLZ3 | DTLZ4 | DTLZ5 | DTLZ6 | DTLZ7 |
|---|---|---|---|---|---|---|---|---|---|---|---|---|
| SPREAD | **4.91** | **6.52** | **5.75** | 4.78 | **4.47** | 11.72 | 13.21 | 10.21 | 30.26 | 10.85 | 8.56 | **11.35** |
| MGD+RBF | 0.00 | 4.64 | 4.97 | **5.27** | 4.39 | 10.17 | 12.83 | 10.26 | **35.38** | 10.79 | **10.92** | 11.23 |
| SPREAD (w/o guidance) | 4.68 | 6.16 | 5.48 | 3.30 | 3.86 | **11.95** | **13.57** | **10.61** | **35.38** | **11.98** | 9.37 | 11.04 |

| Method | RE21 | RE22 | RE25 | RE31 | RE32 | RE33 | RE35 | RE36 | RE37 | RE41 | RE42 | RE61 |
|---|---|---|---|---|---|---|---|---|---|---|---|---|
| SPREAD | **4.83** | **5.18** | **5.33** | **11.87** | 11.34 | **13.50** | 10.82 | 10.03 | 8.39 | **23.16** | **22.87** | 201.43 |
| MGD+RBF | 3.01 | 4.84 | 4.89 | 11.44 | **11.38** | 8.87 | 10.53 | 6.54 | 6.78 | 13.56 | 19.21 | **2004.69** |
| SPREAD (w/o guidance) | 4.79 | 5.14 | 5.24 | **11.87** | 11.17 | 12.72 | **10.85** | **10.13** | **8.51** | 21.34 | 22.40 | 314.52 |

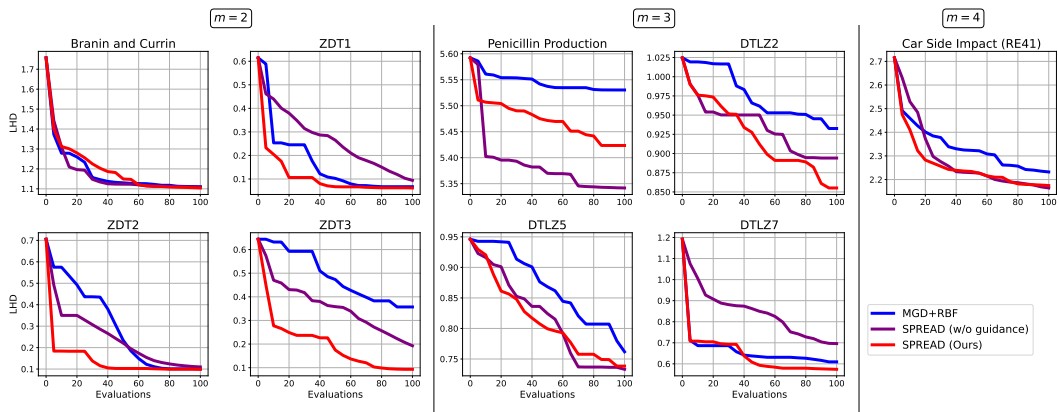

Figure 6: (Bayesian) MGD+RBF baseline and ablation study without guidance. The random seed is fixed to 1000.

**Approximate Pareto Fronts**   To provide a complete view of the results in Tables 1 and 2, Figure 7 illustrates the approximate Pareto optimal points obtained by the different methods on four synthetic and four real-world problems. SPREAD provides broader coverage of the Pareto fronts, particularly on the real-world problems.

**Runtime Scaling with Decision-Space Dimensionality.**   To analyze the runtime scaling with decision space dimensionality, we perform an ablation study on DTLZ4 in the online setting for $(d = 10, 20, 30, 40)$. As shown in Figure 8, the computational cost of SPREAD remains lower than that of PMGDA and higher than that of the other baselines. However, SPREAD delivers better hypervolume and $\Delta$-spread performance than all baselines as the number of decision variables increases.

**Ablation Study on $\nu_t$**   To assess the impact of the parameter $\nu_t$ (equation 13) on the performance of SPREAD, we conduct an ablation study with values ranging from 0 to 100. As shown in Figure 9, setting $\nu_t = 0$ provides a poor trade-off between convergence and diversity. In the bi-objective ZDT2 problem, a lower positive value ($\nu_t = 0.5$) yields both better convergence and good diversity, while in the 4-objective RE41 problem, a higher value ($\nu_t = 50$) achieves a more favorable balance. For the 3-objective DTLZ4 problem, smaller values of $\nu_t$ improve diversity at the expense of convergence, whereas larger values reduce diversity but improve hypervolume. Overall, $\nu_t$ is a key parameter for balancing convergence and diversity in SPREAD, with moderate positive values typically yielding the best performance. In our experiments, we set $\nu_t = 10$

**Ablation Study on $\rho$**   The adaptive perturbation added to the main direction in equation 14 is scaled by $0 < \rho < 1$, which controls its magnitude. We evaluate its effect on SPREAD's performance in Figure 10. The results suggest that when the number of objectives is small (e.g., $m = 2$), lower values of $\rho$ yield good performance, whereas problems with more objectives benefit from moderate or larger values of $\rho$.

**Ablation Study on the Number of Blocks $L$**   Figure 11 shows the performance of SPREAD as the number of blocks increases on ZDT2 ($m = 2$), DTLZ4 ($m = 3$), and RE41 ($m = 4$). The results indicate that a larger number of blocks does not necessarily improve performance, even on problems with more objectives, compared to moderate values such as $L = 2$.

**Effect of the Number of Main-Direction Optimization Steps.**   We performed an ablation on the number of main direction optimization steps $(5, 10, 25, 50)$ on ZDT2, DTLZ4, and RE41. The results are shown in Figure 12. The runtime plot shows that increasing the number of inner steps increases the cost linearly, but the cost remains modest compared to a full DDPM denoising pass. Regarding performance, the hypervolume and $\Delta$-spread curves do not show a consistent monotonic trend across problems: on RE41, performance improves with more steps, while on DTLZ4, the best values appear at smaller step counts. Importantly, all curves remain within a comparable range across the tested step

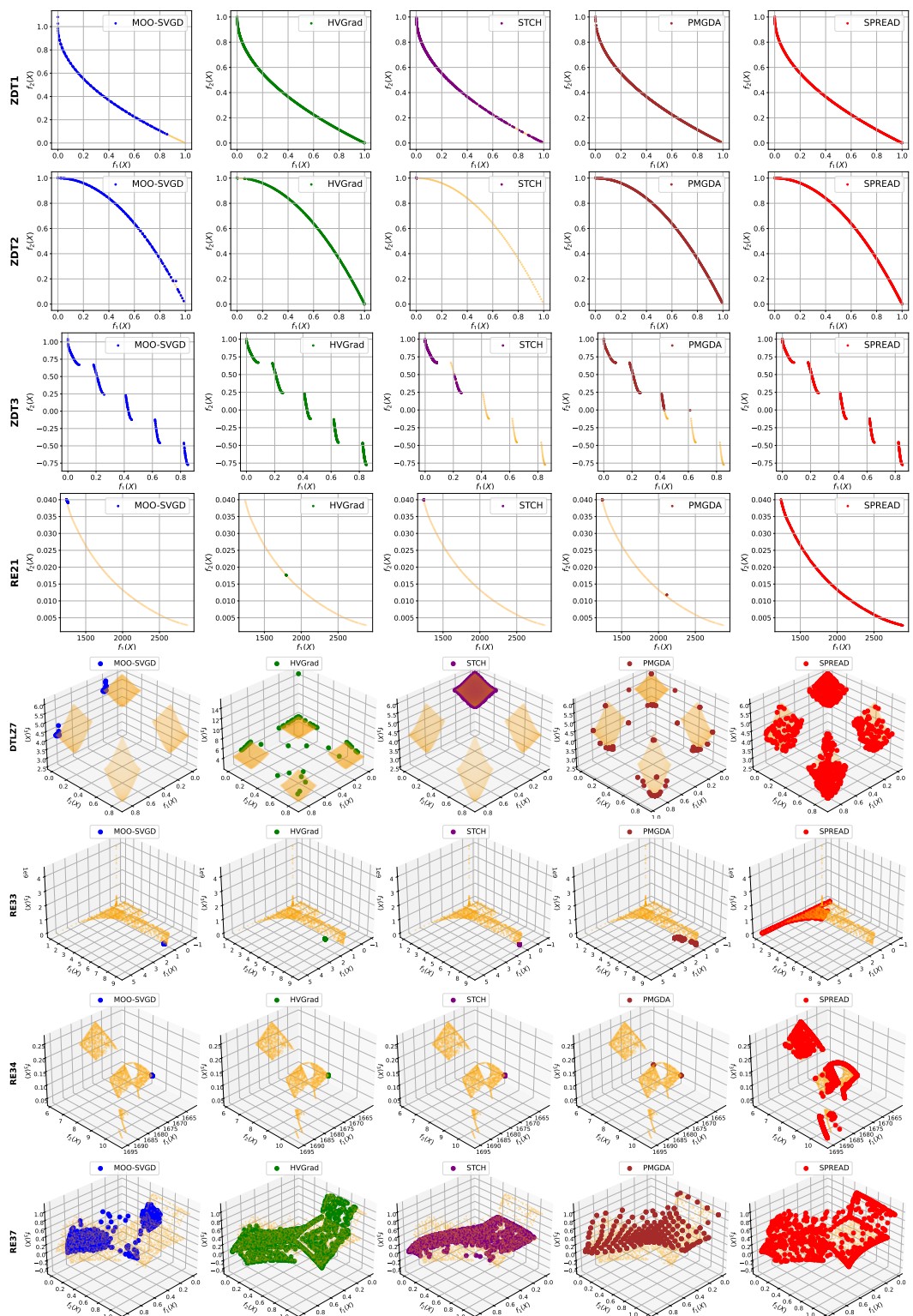

Figure 7: **Approximate Pareto optimal points for multiple benchmark problems.** Solutions from 5 independent runs are merged, and the non-dominated points are shown.

counts. These observations suggest that using a small fixed number of steps (e.g., $5 - 25$) is practical and that the method does not appear particularly sensitive to full convergence of the inner update.

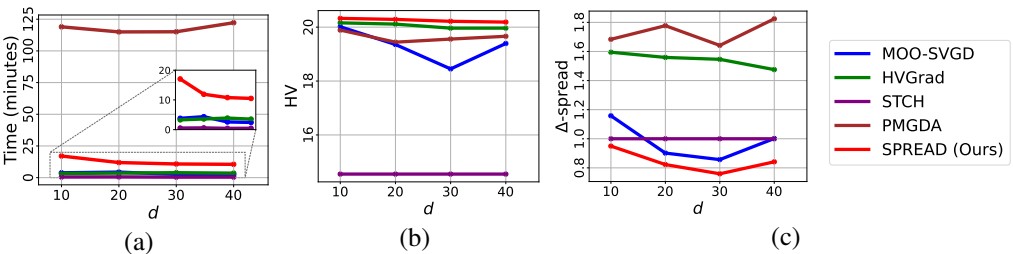

Figure 8: Computational time, hypervolume, and $\Delta$-spread, respectively, as the decision-space dimensionality increases (DTLZ4).

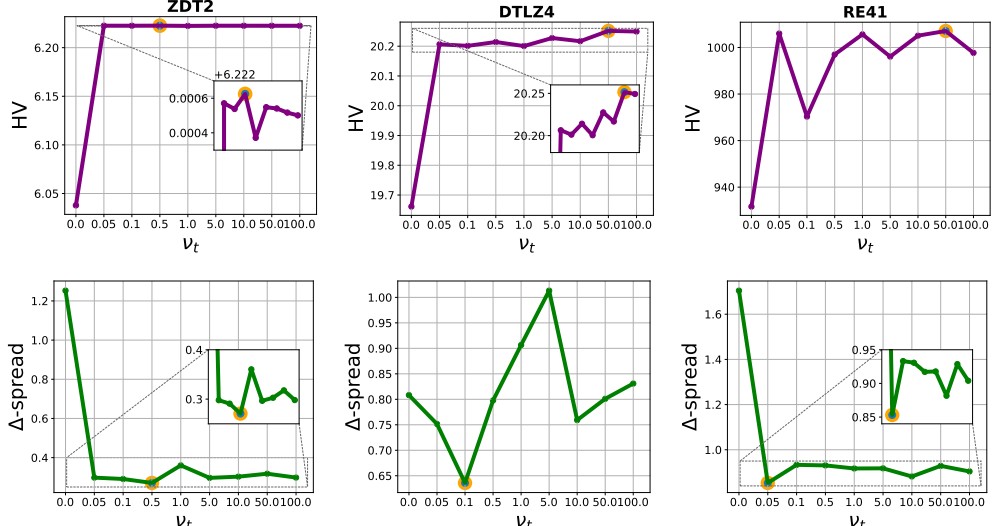

Figure 9: Ablation study on $\nu_t$, evaluated on ZDT2 ($m = 2$), DTLZ4 ($m = 3$), and RE41 ($m = 4$).

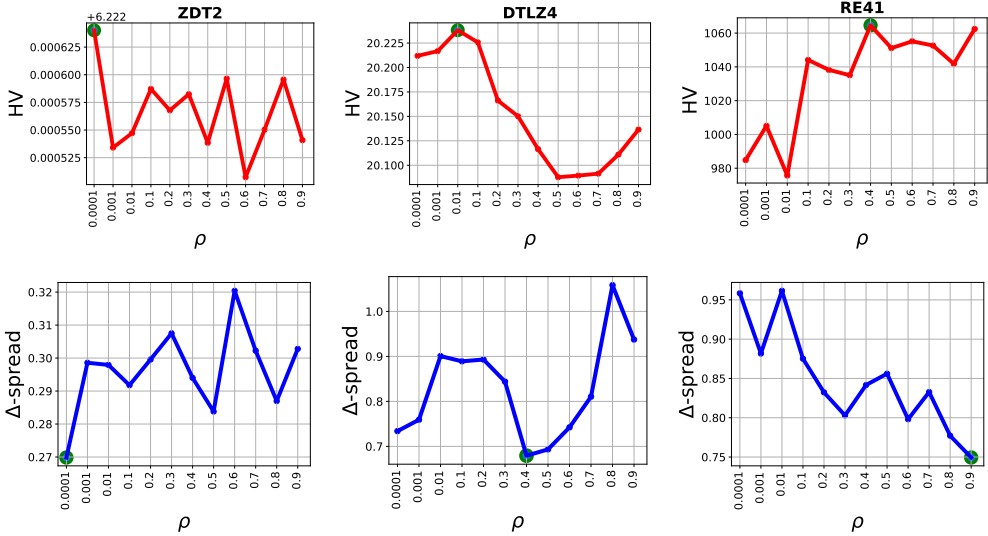

Figure 10: Ablation study on the perturbation scaling factor $\rho$, evaluated on ZDT2 ($m = 2$), DTLZ4 ($m = 3$), and RE41 ($m = 4$).

**Ablation with explicit shift conditioning.** In Table 11, we report the ablation results for explicit shift conditioning in the online setting. For each training sample $\mathbf{x}^i$, the condition is defined as $\mathbf{c} = (\mathbf{F}(\mathbf{x}^i), \Xi)$, instead of the implicit form $\mathbf{c} = \mathbf{F}(\mathbf{x}^i) + \Xi$. This requires embedding $\mathbf{F}(\mathbf{x}^i)$

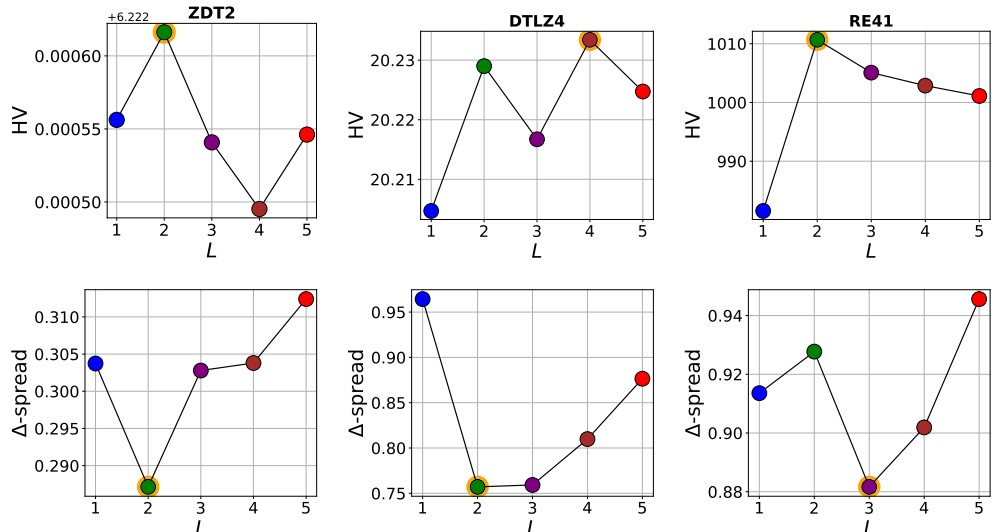

Figure 11: Ablation study on the number of blocks $L \in \{1, 2, \ldots, 5\}$ in DiT-MOO, evaluated on ZDT2 ($m = 2$), DTLZ4 ($m = 3$), and RE41 ($m = 4$).

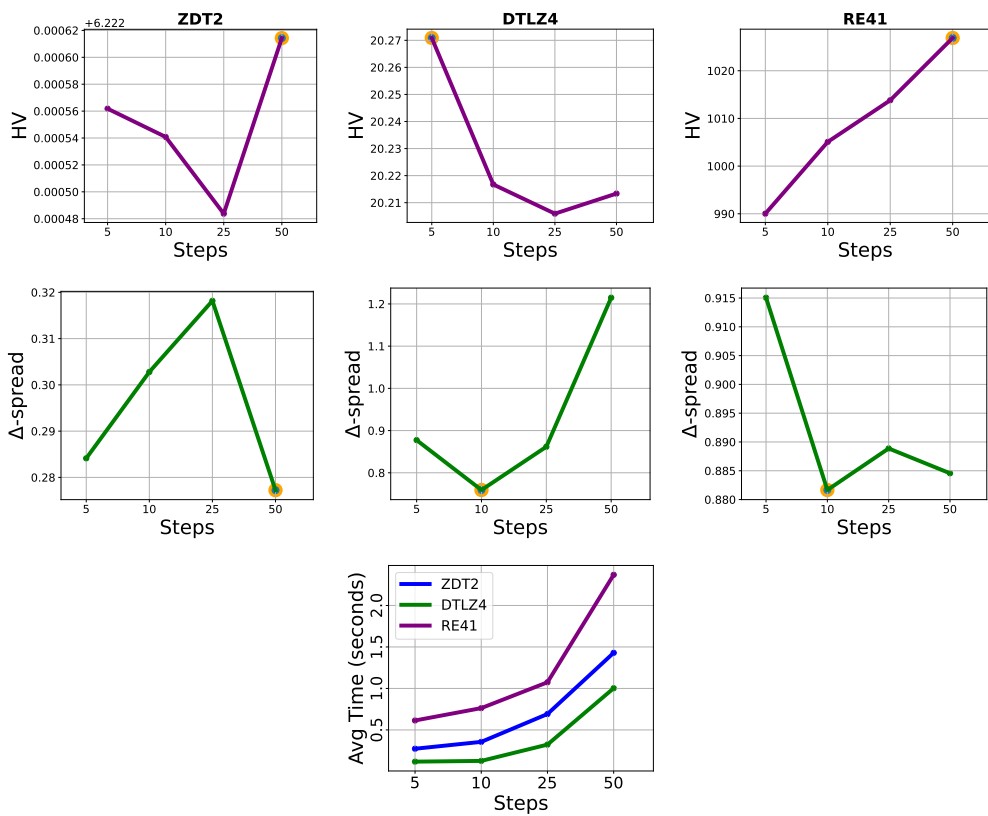

Figure 12: Ablation study on the number of optimization steps in equation 13, evaluated on ZDT2 ($m = 2$), DTLZ4 ($m = 3$), and RE41 ($m = 4$).

and $\Xi$ separately and concatenating their embeddings with the time embedding. Consequently, the conditioning input to the multi-head cross-attention module has shape $(n, 3, e)$, while the output retains the shape $(n, 1, e)$ as in Figure 1. During sampling, we set the condition to $\mathbf{c} = (\mathbf{F}(\mathbf{x}^i), 0)$.

Overall, the results indicate that the implicit shift conditioning used in SPREAD slightly outperforms explicit shift conditioning in terms of hypervolume on most problems, with the difference being especially notable on the 4-objective RE41 problem. Explicit conditioning, however, improves solution diversity on the ZDT problems and achieves higher hypervolume on RE33 and RE34. Nevertheless, the theoretical improvement guarantee established for implicit shift conditioning in Theorem 1 does not directly carry over to the explicit variant.

Table 11: Ablation study with explicit shift conditioning. The random seed is fixed to 1000, and the best results are highlighted in bold.

| Problem | SPREAD | | SPREAD (explicit) | |
|---|---|---|---|---|
| | HV | $\Delta$-spread | HV | $\Delta$-spread |
| ZDT1 | **5.72** | 0.32 | **5.72** | **0.29** |
| ZDT2 | **6.22** | 0.30 | **6.22** | **0.28** |
| ZDT3 | **6.10** | 0.53 | **6.10** | **0.50** |
| RE21 | **70.11** | **0.41** | 70.10 | 0.43 |
| DTLZ2 | **22.92** | 0.92 | 22.91 | 0.94 |
| DTLZ4 | **20.22** | **0.76** | 20.22 | **0.76** |
| DTLZ7 | **18.08** | **0.61** | 18.07 | 0.69 |
| RE33 | 135.26 | **0.97** | 135.29 | 1.03 |
| RE34 | 242.69 | 0.93 | **242.91** | **0.84** |
| RE37 | **1.42** | 0.81 | **1.42** | **0.73** |
| RE41 | **1005.08** | **0.88** | 968.46 | 0.99 |

**Training Cost and Feasibility of Transfer Learning Across Related Problems.** We acknowledge that training a separate diffusion model for each problem introduces additional computational overhead. However, when multiple problems share the same decision space and number of objectives, transfer learning becomes feasible and can substantially reduce the training effort in both online and offline settings. In this experiment, we perform transfer learning by training a single diffusion model on groups of related problems. Two groups are considered in both the online and offline settings:

- Online setting: (ZDT1-3) and (DTLZ2, DTLZ4, DTLZ7)

- Offline setting: (ZDT1-3) and (DTLZ2-7)

Tables 12 and 13 present the results. In the online setting, transfer learning achieves performance comparable to training individual models, with only one exception (DTLZ2) where the hypervolume is slightly lower. In the offline setting, transfer learning even improves performance on most problems. Overall, these experiments suggest that transfer learning is a practical strategy for reducing training cost with minimal risk of performance degradation.

Table 12: (Online) Transfer learning results. The random seed is fixed to 1000, and the best results are highlighted in bold.

| Problem | SPREAD | | SPREAD (transfer) | |
|---|---|---|---|---|
| | HV | $\Delta$-spread | HV | $\Delta$-spread |
| ZDT1 | **5.72** | **0.32** | **5.72** | **0.32** |
| ZDT2 | **6.22** | 0.30 | **6.22** | **0.29** |
| ZDT3 | **6.10** | **0.53** | **6.10** | **0.53** |
| DTLZ2 | **22.92** | 0.92 | 22.91 | **0.90** |
| DTLZ4 | **20.22** | **0.76** | **20.22** | 0.79 |
| DTLZ7 | **18.08** | **0.61** | **18.08** | 0.70 |

Table 13: (Offline) Transfer learning results. The random seed is fixed to 1000, and the best results are highlighted in bold.

| Method | ZDT1 | ZDT2 | ZDT3 | DTLZ2 | DTLZ3 | DTLZ4 | DTLZ5 | DTLZ6 | DTLZ7 |
|---|---|---|---|---|---|---|---|---|---|
| SPREAD | **4.91** | **6.52** | 5.75 | **13.21** | 10.21 | 30.26 | **10.85** | 8.56 | **11.35** |
| SPREAD (transfer) | **4.91** | **6.52** | **5.85** | 13.17 | **10.22** | **31.09** | **10.85** | **9.70** | **11.35** |

Table 14: (Offline) Hypervolume results of synthetic functions averaged over 5 independent runs.

| HV (↑) | m = 2 | | | | | m = 3 | | | | | | |
|---|---|---|---|---|---|---|---|---|---|---|---|---|
| Method | ZDT1 | ZDT2 | ZDT3 | ZDT4 | ZDT6 | DTLZ1 | DTLZ2 | DTLZ3 | DTLZ4 | DTLZ5 | DTLZ6 | DTLZ7 |
| $\mathcal{D}$(best) | 4.17 | 4.67 | 5.15 | 5.45 | 4.61 | 10.60 | 9.91 | 10.00 | 10.76 | 9.35 | 8.88 | 8.56 |
| MM | 4.81 ± 0.02 | 5.57 ± 0.07 | 5.48 ± 0.21 | 5.03 ± 0.19 | 4.78 ± 0.01 | 10.64 ± 0.01 | 9.03 ± 0.80 | 10.58 ± 0.03 | 7.66 ± 1.30 | 7.65 ± 1.39 | 9.58 ± 0.31 | 10.61 ± 0.16 |
| MM-COM | 4.52 ± 0.02 | 4.99 ± 0.12 | 5.49 ± 0.07 | 5.10 ± 0.08 | 4.41 ± 0.21 | 10.64 ± 0.01 | 8.99 ± 0.97 | 10.27 ± 0.37 | 9.72 ± 0.39 | 9.44 ± 0.41 | 9.37 ± 0.35 | 10.09 ± 0.36 |
| MM-IOM | 4.68 ± 0.12 | 5.45 ± 0.11 | 5.61 ± 0.06 | 4.99 ± 0.21 | 4.75 ± 0.01 | 10.64 ± 0.01 | 10.10 ± 0.27 | 10.24 ± 0.13 | 10.03 ± 0.53 | 9.77 ± 0.18 | 9.30 ± 0.31 | 10.60 ± 0.05 |
| MM-ICT | 4.82 ± 0.01 | 5.58 ± 0.01 | 5.59 ± 0.06 | 4.63 ± 0.43 | 4.75 ± 0.01 | 10.64 ± 0.01 | 8.68 ± 0.88 | 10.25 ± 0.42 | 10.33 ± 0.24 | 9.25 ± 0.28 | 9.10 ± 1.16 | 10.29 ± 0.05 |
| MM-RoMA | 4.84 ± 0.01 | 5.43 ± 0.35 | **5.89 ± 0.04** | 4.13 ± 0.11 | 1.71 ± 0.10 | 10.64 ± 0.01 | 10.04 ± 0.05 | 10.61 ± 0.03 | 9.25 ± 0.11 | 8.71 ± 0.47 | 9.84 ± 0.25 | 10.53 ± 0.04 |
| MM-TriMentoring | 4.64 ± 0.10 | 5.22 ± 0.11 | 5.16 ± 0.04 | **5.12 ± 0.12** | 2.61 ± 0.01 | 10.64 ± 0.01 | 9.39 ± 0.35 | 10.48 ± 0.12 | 10.21 ± 0.06 | 7.69 ± 1.03 | 9.00 ± 0.48 | 10.12 ± 0.09 |
| MH | 4.80 ± 0.03 | 5.57 ± 0.07 | 5.58 ± 0.20 | 4.59 ± 0.26 | 4.78 ± 0.01 | 10.51 ± 0.23 | 9.03 ± 0.56 | 10.48 ± 0.23 | 6.73 ± 1.40 | 8.41 ± 0.15 | 8.72 ± 1.07 | 10.66 ± 0.09 |
| MH-PcGrad | 4.84 ± 0.01 | 5.55 ± 0.11 | 5.51 ± 0.03 | 3.68 ± 0.70 | 4.67 ± 0.10 | 10.64 ± 0.01 | 9.64 ± 0.33 | 10.55 ± 0.12 | 9.95 ± 1.13 | 9.02 ± 0.24 | 9.90 ± 0.25 | 10.61 ± 0.03 |
| MH-GradNorm | 4.63 ± 0.15 | 5.37 ± 0.17 | 5.54 ± 0.20 | 3.28 ± 0.90 | 3.81 ± 1.20 | 10.64 ± 0.01 | 8.86 ± 1.27 | 10.26 ± 0.28 | 7.45 ± 0.75 | 7.87 ± 1.06 | 8.16 ± 2.21 | 10.31 ± 0.22 |
| ParetoFlow | 4.23 ± 0.04 | 5.65 ± 0.11 | 5.29 ± 0.14 | 5.00 ± 0.22 | 4.48 ± 0.11 | 10.60 ± 0.02 | 10.13 ± 0.16 | 10.41 ± 0.09 | 10.29 ± 0.17 | 9.65 ± 0.23 | 9.25 ± 0.43 | 8.94 ± 0.18 |
| PGD-MOO | 4.41 ± 0.08 | 5.33 ± 0.05 | 5.54 ± 0.10 | 5.02 ± 0.03 | 4.82 ± 0.01 | 10.65 ± 0.01 | 10.55 ± 0.01 | **10.63 ± 0.01** | 10.64 ± 0.01 | 10.06 ± 0.02 | 10.14 ± 0.01 | 9.70 ± 0.18 |
| **SPREAD** | **4.89 ± 0.02** | **6.52 ± 0.00** | 5.82 ± 0.04 | 4.90 ± 0.13 | 4.51 ± 0.04 | **11.46 ± 0.13** | **13.27 ± 0.02** | 10.23 ± 0.01 | **31.19 ± 1.10** | **10.85 ± 0.00** | 9.56 ± 0.42 | **11.35 ± 0.00** |

Table 15: (Offline) Hypervolume results of real-world tasks averaged over 5 independent runs.

| HV (↑) | m = 2 | | | m = 3 | | | | | | m = 4 | | m = 6 |
|---|---|---|---|---|---|---|---|---|---|---|---|---|
| Method | RE21 | RE22 | RE25 | RE31 | RE32 | RE33 | RE35 | RE36 | RE37 | RE41 | RE42 | RE61 |
| $\mathcal{D}$(best) | 4.10 | 4.78 | 4.79 | 10.6 | 10.56 | 10.56 | 10.08 | 7.61 | 5.57 | 18.27 | 14.52 | 97.49 |
| MM | 4.60 ± 0.00 | 4.84 ± 0.00 | 4.63 ± 0.25 | 10.65 ± 0.00 | 10.62 ± 0.02 | 10.62 ± 0.00 | 10.55 ± 0.01 | 10.24 ± 0.03 | 6.73 ± 0.03 | 20.77 ± 0.08 | 22.59 ± 0.11 | 108.96 ± 0.06 |
| MM-COM | 4.38 ± 0.09 | 4.84 ± 0.00 | 4.83 ± 0.01 | 10.64 ± 0.01 | 10.64 ± 0.01 | 10.61 ± 0.00 | 10.55 ± 0.02 | 9.82 ± 0.35 | 6.35 ± 0.10 | 20.37 ± 0.06 | 17.44 ± 0.71 | 107.99 ± 0.48 |
| MM-IOM | 4.58 ± 0.02 | 4.84 ± 0.00 | 4.83 ± 0.01 | 10.65 ± 0.00 | 10.65 ± 0.00 | 10.62 ± 0.00 | 10.57 ± 0.01 | **10.29 ± 0.04** | 6.71 ± 0.02 | 20.66 ± 0.05 | 22.43 ± 0.10 | 107.71 ± 0.50 |
| MM-ICT | 4.60 ± 0.00 | 4.84 ± 0.00 | 4.84 ± 0.00 | 10.65 ± 0.00 | 10.64 ± 0.00 | 10.62 ± 0.00 | 10.50 ± 0.01 | 10.29 ± 0.03 | 6.73 ± 0.00 | 20.58 ± 0.04 | 22.27 ± 0.15 | 108.68 ± 0.27 |
| MM-RoMA | 4.57 ± 0.00 | 4.61 ± 0.51 | 4.83 ± 0.01 | 10.64 ± 0.01 | 10.64 ± 0.00 | 10.58 ± 0.03 | 10.53 ± 0.03 | 9.72 ± 0.28 | 6.67 ± 0.02 | 20.39 ± 0.09 | 21.41 ± 0.37 | 108.47 ± 0.28 |
| MM-TriMentoring | 4.60 ± 0.00 | 4.84 ± 0.00 | 4.84 ± 0.00 | 10.65 ± 0.00 | 10.62 ± 0.01 | 10.60 ± 0.01 | 10.59 ± 0.00 | 9.64 ± 1.42 | 6.73 ± 0.01 | 20.68 ± 0.04 | 21.60 ± 0.19 | 108.61 ± 0.29 |
| MH | 4.60 ± 0.00 | 4.84 ± 0.00 | 4.74 ± 0.20 | 10.65 ± 0.00 | 10.60 ± 0.05 | 10.62 ± 0.00 | 10.59 ± 0.00 | 10.49 ± 0.07 | 6.67 ± 0.05 | 20.62 ± 0.11 | 22.38 ± 0.35 | 108.92 ± 0.22 |
| MH-PcGrad | 4.59 ± 0.01 | 4.73 ± 0.36 | 4.78 ± 0.14 | 10.64 ± 0.01 | 10.63 ± 0.01 | 10.59 ± 0.03 | 10.51 ± 0.05 | 10.17 ± 0.08 | 6.68 ± 0.06 | 20.66 ± 0.10 | 22.57 ± 0.26 | 108.54 ± 0.23 |
| MH-GradNorm | 4.28 ± 0.39 | 4.70 ± 0.44 | 4.52 ± 0.50 | 10.60 ± 0.10 | 10.54 ± 0.15 | 10.03 ± 1.50 | 9.76 ± 1.30 | 9.67 ± 0.43 | 5.67 ± 1.41 | 17.06 ± 3.82 | 18.77 ± 2.99 | 108.01 ± 1.00 |
| ParetoFlow | 4.20 ± 0.17 | 4.86 ± 0.01 | 4.84 ± 0.00 | 10.66 ± 0.12 | 10.61 ± 0.00 | 10.75 ± 0.20 | **11.12 ± 0.02** | 8.42 ± 0.35 | 6.55 ± 0.59 | 19.41 ± 0.92 | 20.35 ± 5.31 | 107.10 ± 6.96 |
| PGD-MOO | 4.46 ± 0.03 | 4.84 ± 0.00 | 4.84 ± 0.00 | 10.60 ± 0.01 | 10.65 ± 0.00 | 10.51 ± 0.04 | 10.32 ± 0.10 | 9.37 ± 0.17 | 6.13 ± 0.12 | 19.31 ± 0.46 | 19.01 ± 0.68 | 105.02 ± 1.14 |
| **SPREAD** | **4.83 ± 0.00** | **5.18 ± 0.00** | **5.33 ± 0.06** | **11.87 ± 0.00** | **11.27 ± 0.17** | **13.50 ± 0.02** | 10.78 ± 0.03 | 9.55 ± 0.24 | **8.37 ± 0.06** | **22.29 ± 0.26** | **23.18 ± 0.96** | **251.81 ± 49.97** |

Table 16: (Offline: comparison with evolutionary algorithms) Hypervolume results of synthetic functions. Best values are highlighted in **bold**.

| HV (↑) | m = 2 | | | | | m = 3 | | | | | | |
|---|---|---|---|---|---|---|---|---|---|---|---|---|
| Method | ZDT1 | ZDT2 | ZDT3 | ZDT4 | ZDT6 | DTLZ1 | DTLZ2 | DTLZ3 | DTLZ4 | DTLZ5 | DTLZ6 | DTLZ7 |
| NSGA-III | 4.85 ± 0.00 | 5.70 ± 0.00 | 5.68 ± 0.02 | 4.50 ± 0.05 | 4.76 ± 0.01 | 10.65 ± 0.00 | 7.88 ± 1.07 | **10.56 ± 0.08** | 7.01 ± 0.12 | 8.98 ± 0.13 | 9.36 ± 0.32 | 10.79 ± 0.00 |
| MOEA/D | 4.85 ± 0.00 | 5.69 ± 0.00 | 5.65 ± 0.06 | 4.38 ± 0.16 | **4.79 ± 0.00** | 10.17 ± 0.42 | 7.00 ± 0.49 | 9.82 ± 0.33 | 7.94 ± 0.80 | 7.30 ± 0.88 | 6.34 ± 0.33 | 10.43 ± 0.00 |
| **SPREAD** | **4.89 ± 0.02** | **6.52 ± 0.00** | **5.82 ± 0.04** | **4.90 ± 0.13** | 4.51 ± 0.04 | **11.46 ± 0.13** | **13.27 ± 0.02** | 10.23 ± 0.01 | **31.19 ± 1.10** | **10.85 ± 0.00** | **9.56 ± 0.42** | **11.35 ± 0.00** |

Table 17: (Offline: comparison with evolutionary algorithms) Hypervolume results of real-world tasks. Best values are highlighted in **bold**.

| HV (↑) | m = 2 | | | m = 3 | | | | | | m = 4 | | m = 6 |
|---|---|---|---|---|---|---|---|---|---|---|---|---|
| Method | RE21 | RE22 | RE25 | RE31 | RE32 | RE33 | RE35 | RE36 | RE37 | RE41 | RE42 | RE61 |
| NSGA-III | 4.57 ± 0.00 | 4.84 ± 0.00 | 4.35 ± 0.00 | 10.65 ± 0.00 | 10.63 ± 0.01 | 10.61 ± 0.01 | 10.49 ± 0.00 | **9.98 ± 0.07** | 6.69 ± 0.01 | 20.77 ± 0.01 | 22.30 ± 0.19 | 108.94 ± 0.10 |
| MOEA/D | 4.57 ± 0.00 | 4.84 ± 0.00 | 4.35 ± 0.00 | 10.25 ± 0.04 | 10.63 ± 0.00 | 10.58 ± 0.01 | 10.35 ± 0.10 | 9.83 ± 0.08 | 6.66 ± 0.00 | 21.09 ± 0.00 | 22.09 ± 0.01 | NA |
| **SPREAD** | **4.83 ± 0.00** | **5.18 ± 0.00** | **5.33 ± 0.06** | **11.87 ± 0.00** | **11.27 ± 0.17** | **13.50 ± 0.02** | **10.78 ± 0.03** | 9.55 ± 0.24 | **8.37 ± 0.06** | **22.29 ± 0.26** | **23.18 ± 0.96** | **251.81 ± 49.97** |

**Hypervolume Results in the Offline Setting** As mentioned in Section 5.2, Table 4 reports the average rank results. For reference, we provide here the corresponding individual hypervolume results: Table 14 summarizes the results for synthetic problems, while Table 15 reports the results for real-world tasks. For each problem, the overall best method is shown in **bold**, and the best generative approach is highlighted in  light gray . In addition, we evaluate the ability of the well-known evolutionary algorithms NSGA-III and MOEA/D, originally designed for the online setting, on offline MOO tasks. We use the `pymoo` (Blank & Deb, 2020) implementation, with the evaluation adapted to rely on pretrained proxy models instead of the true objective functions. The label "NA" denotes runs that failed due to memory exhaustion when handling many objectives. As shown in Tables 16 and 17, these algorithms struggle to adapt to the offline setting, indicating that state-of-the-art online MOO methods are not necessarily suitable for resource-constrained domains. By contrast, our SPREAD framework provides this flexibility.

**Ablation Study in the MOBO Setting** Figure 13 complements the results in Section 5.3 by comparing the SPREAD and CDM-PSL generative methods without SBX (step 8, Algorithm 3). The figure highlights that SPREAD achieves superior performance compared to CDM-PSL under this setting. Interestingly, on the 4-objective Car Side Impact problem, SPREAD achieves a better convergence rate without SBX than with it.

**Many-Objective Experiments** ($m = 10$)**.** We consider the DTLZ problem family (online setting), which is flexible with respect to the number of objectives. Since HVGrad is a hypervolume-maximization method, it cannot be applied in this setting because hypervolume computation becomes

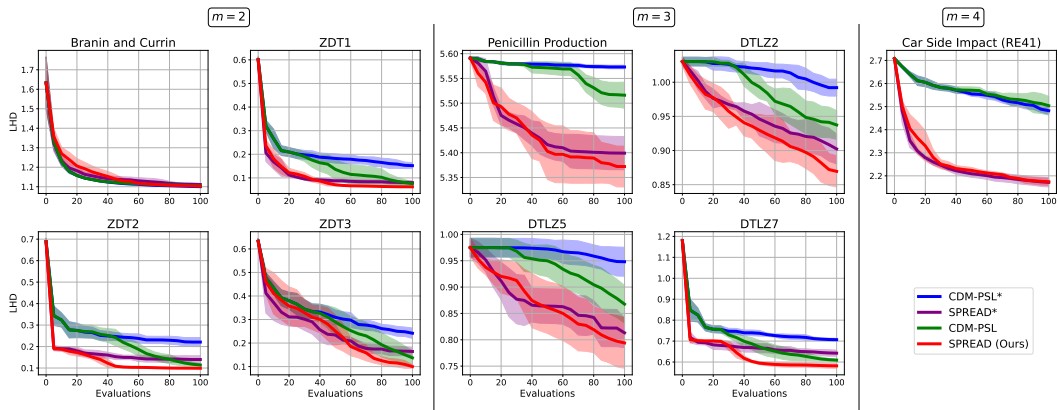

Figure 13: (Bayesian) Ablation study on the local-optima escaping technique at step 8 of Algorithm 3. SPREAD is compared against CDM-PSL, where variants without step 8 are marked with an asterisk (*).

prohibitively expensive in many-objective optimization, and we were therefore unable to run HVGrad. We therefore exclude it from this experiment. To evaluate performance, we use an approximate hyper-volume calculation based on the *hvwfg*(Walking Fish Group algorithm (While & Bradstreet, 2012)) implementation from the `pygmo` library (Biscani & Izzo, 2020). As shown in Figure 14, SPREAD outperforms the remaining three baselines on DTLZ4 and DTLZ7, while PMGDA achieves the best performance on DTLZ2. Overall, these results demonstrate that SPREAD remains competitive in the many-objective setting and scales favorably to higher numbers of objectives.

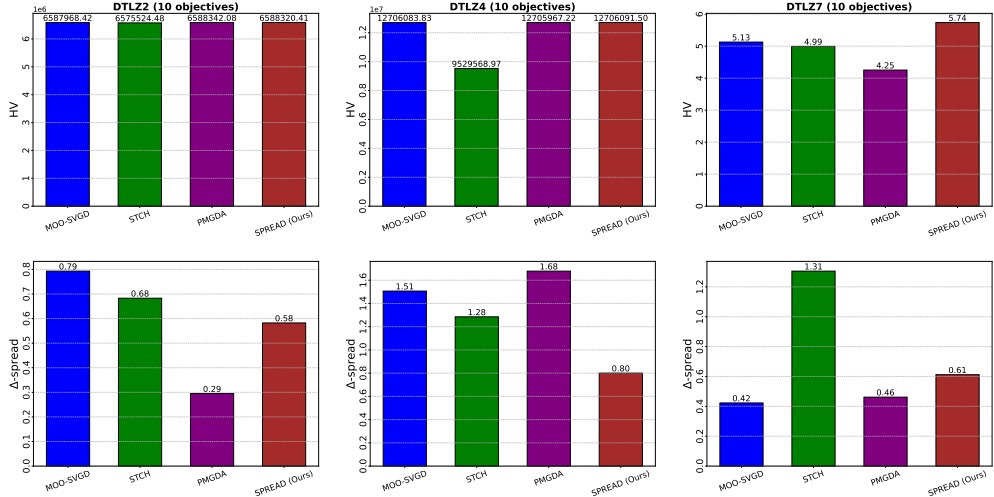

Figure 14: Performance comparison for many-objective DTLZ problems with $m = 10$. We report hypervolume (higher is better) and $\Delta$-spread (lower is better). The random seed is fixed to 1000.

# E EXTENDED RELATED WORK

To complement the discussion in Section 2, we provide a broader overview of related work, ranging from diffusion-based approaches for single-objective black-box optimization (BBO) to surrogate-assisted methods for multi-objective optimization.

**Diffusion Models as Data-Driven Samplers for Black-Box Optimization** In the single-objective, offline setting, Krishnamoorthy et al. (2023) introduced Denoising Diffusion Optimization Models

(DDOM), which learn a conditional generative model over designs given target values and, at test time, employ guidance to sample high-reward candidates. This approach highlights the potential of inverse modeling without relying on explicit surrogates. Beyond the offline setting, Diffusion-BBO (Wu et al., 2024) extends this idea to the online regime by scoring in the objective space through an uncertainty-aware acquisition function and then conditionally sampling designs, with theoretical and empirical results showing sample-efficient improvements over BO baselines. Parallel efforts broaden the scope of data-driven BBO with diffusion by training reward-directed conditional models that combine large unlabeled datasets with small labeled sets and provide sub-optimality guarantees, or by constraining sampling to learned data manifolds to enforce feasibility, both yielding improvements across black-box optimization tasks (Li et al., 2024b). Together, these works highlight the potential of diffusion models to reformulate single-objective black-box optimization as a generative sampling task, paving the way for extensions to multi-objective settings.

**Surrogate-Assisted Multi-Objective Optimization**    Surrogate techniques have long been combined with multi-objective optimization to reduce the cost of expensive evaluations by approximating the true objectives, either globally or locally. Deb et al. (2020) offer a broad taxonomy of surrogate modeling strategies and propose an adaptive switching scheme (ASM) that cycles among different surrogate types, demonstrating that ASM often outperforms any individual surrogate model. Without requiring explicit gradients, Berkemeier & Peitz (2021) introduce a derivative-free trust-region descent method for multi-objective problems, which builds radial basis function surrogates within each local region and proves convergence to Pareto-critical points. In practical engineering settings, surrogate-assisted MOO has been applied to optimize permanent magnet synchronous motors using neural networks, Kriging, or support vector regression under small sample regimes (Li et al., 2025b). In the context of multi-objective control problems and PDE-constrained systems, Peitz & Dellnitz (2018b) survey how surrogate modeling or reduced-order models help accelerate decision making or feedback control under real-time constraints. In reservoir modeling and well control, the MOO-SESA framework of Wang et al. (2024) combines a selective ensemble of SVR surrogates with NSGA-II to strike a balance between surrogate robustness and multi-objective accuracy, yielding faster convergence and more reliable Pareto fronts in benchmark reservoir. Overall, surrogate-assisted methods provide the foundation for offline and Bayesian multi-objective optimization, where surrogates not only reduce evaluation costs but also serve as probabilistic models to guide exploration and exploitation under uncertainty.

