# OpenReview forum: "SPREAD: Sampling-based Pareto front Refinement via Efficient Adaptive Diffusion"
_ICLR.cc/2026/Conference — ICLR 2026 Poster_

### Official Review · Reviewer_rKVB · 2025-10-24

**Soundness:** 2
**Presentation:** 2
**Contribution:** 3
**Rating:** 6
**Confidence:** 2

**Summary:**

This paper introduces SPREAD, a generative framework for multi-objective optimization (MOO) built upon Denoising Diffusion Probabilistic Models (DDPMs). The central idea is to learn a conditional diffusion process over the decision space and refine candidate solutions during reverse diffusion via an adaptive guidance mechanism. This mechanism combines a Multiple Gradient Descent (MGD)-inspired update for efficient convergence and a Gaussian RBF-based repulsion term for maintaining diversity. The paper clearly motivates the use of a generative, diffusion-based formulation by highlighting the limitations of conventional MOO approaches. The main contribution lies in unifying these components into a single conditional diffusion process, applicable to online, offline, and Bayesian MOO settings. Extensive experiments on synthetic and real-world benchmarks demonstrate that SPREAD achieves competitive or superior performance in hypervolume and diversity, particularly as the number of objectives increases.

**Strengths:**

- The paper introduces a fresh perspective on MOO by leveraging diffusion models for iterative refinement toward the Pareto front
- The combination of MGD-based guidance and RBF repulsion is both intuitive and well-motivated, yielding a good balance between convergence and diversity.
- The theoretical results (Theorems 1 and 2) are rigorous, supporting the conditioning and convergence properties.
- SPREAD consistently outperforms or matches state-of-the-art methods in hypervolume and diversity metrics.
- Ablation studies (Table 3, Figs. 6–8) are particularly informative, showing that each component—MGD alignment, repulsion, and perturbation—contributes meaningfully to performance.

**Weaknesses:**

- The notation is overly dense, with multiple primes and sub-/superscripts that reduce readability.
- The framework involves training a diffusion transformer and solving an optimization (Eq. 13) at every sampling step. Although Figure 2 shows SPREAD is faster than PMGDA, it remains slower than lighter baselines such as MOO-SVGD or HVGrad. Particularly, solving Eq. 13 not only depends on the data size $n$ but also the decision-space dimensionality. Some analysis of runtime scaling with decision-space dimensionality would strengthen the discussion.
- SPREAD introduces two hyperparameters, $\nu_t$ and $\rho$. While $\nu_t = 10$ is reported as the default, the default for $\rho$ is unspecified. Given the cost of expensive optimization runs, hyperparameter tuning can be prohibitive. Moreover, Figures 6 and 7 indicate that performance is somewhat sensitive to these values, suggesting that robustness should be discussed in greater detail.
- The largest tested problem (RE61) has $m = 6$ objectives. In combination with the weakness of *hyperparameter sensitivity*, it would be valuable to evaluate SPREAD's performance in higher-dimensional settings (e.g., $m \geq 10$) to assess effectiveness of SPREAD.

**Questions:**

Kindly address weaknesses.

---

> ### Author Response · Authors · 2025-11-21
>
> We appreciate the thoughtful comments and valuable suggestions of the reviewer.
>
> ## **Addressing the Weaknesses**
>
> - **Notation Simplifications for Improved Readability.**
> To improve readability, we have simplified the notation used for the MGD and guidance directions. Since these directions are always computed from $\mathbf{X}\_{t}^{'}$, their dependence on the primed variables is unambiguous, and the prime notation is therefore unnecessary. In the revised manuscript, we replace $\mathbf{g}\_{t}^{'i}$, $\mathbf{h}\_{t}^{'i}$, and $\tilde{\mathbf{h}}_{t}^{'i}$ with the cleaner forms $\mathbf{g}\_{t}^{i}$, $\mathbf{h}\_{t}^{i}$, and $\tilde{\mathbf{h}}\_{t}^{i}$, respectively. This change substantially reduces subscript/superscript clutter and improves the overall readability of the equations and algorithmic steps.
>
> - **Runtime Scaling with Decision Space Dimensionality.**
> To analyze the runtime scaling with decision space dimensionality, we perform an ablation study on DTLZ4 in the online setting for $d = 10, 20, 30, 40$. As shown in Figure 8, the computational cost of SPREAD remains lower than that of PMGDA and higher than that of the other baselines. However, SPREAD delivers better hypervolume and $\Delta$-spread performance than all baselines as the number of decision variables increases.
>
> - **Choice of $\rho$ and Sensitivity to Its Value.**
> We thank the reviewer for pointing out that the default choice of the hyperparameter $\rho$ was not explicitly reported. We have now added this information to the revised manuscript. In all experiments, we use fixed values of $\rho$ determined solely by the number of objectives and, in a few cases, by the specific problem instance:
>   - *Online setting:*
>     - $m = 2$: $\rho = 0.9$
>     - $m > 2$: $\rho = 0.001$
>
>   - *Offline setting:*
>     - $m = 2$: $\rho = 0.9$, except for ZDT4 ($\rho = 0.0001$) and ZDT6 ($\rho = 0.1$)
>     - $m > 2$: $\rho = 0.001$, except for DTLZ4 and DTLZ6 ($\rho = 0.0001$)
>
>   - *Bayesian setting:*
>     - $m = 2$: $\rho = 0.9$
>     - $m > 2$: $\rho = 0.01$
>
>   We determined the default $\rho$ values from a few preliminary checks and used them unchanged throughout our experiments. We did not conduct any extensive hyperparameter search. While Figure $7$ (now Figure $10$ in the revised manuscript) shows that SPREAD's performance varies across a wide range of $\rho$ values, the relative ordering with respect to the baselines remains stable: even with non-optimal choices of $\rho$, SPREAD does not exhibit any drastic performance degradation relative to the competing methods. For new problems, we recommend using the default values, which depend only on the number of objectives. We now discuss this robustness and provide the exact defaults in the revised manuscript to make our experimental setup fully transparent.
>
> - **Performance of SPREAD on Many-Objective DTLZ Problems.**
> We have added many-objective experiments $(m=10)$ using the DTLZ family. Because HVGrad relies on hypervolume maximization and hypervolume becomes prohibitively expensive in many-objective optimization, we exclude it from this setting. For evaluation, we use the approximate hypervolume computed via the *hvwfg* implementation in *pygmo*. As shown in Figure 14, SPREAD outperforms the baselines on DTLZ4 and DTLZ7, while PMGDA performs best on DTLZ2. Overall, SPREAD remains competitive in the many-objective scenario.

---

> > ### Comment · Reviewer_rKVB · 2025-11-24
> > **Official Comment by Reviewer rKVB**
> >
> > Thank you for the extensive and detailed answers. They have addressed all of my concerns and the revised paper has been considerably improved. As a result, I am increasing my score.

---

> > > ### Author Response · Authors · 2025-11-24
> > >
> > > We thank Reviewer rKVB for the valuable feedback and careful re-evaluation. We are pleased that the revisions satisfactorily addressed all concerns and appreciate the reviewer's decision to raise the score.

---

### Official Review · Reviewer_CrTm · 2025-10-28

**Soundness:** 2
**Presentation:** 3
**Contribution:** 2
**Rating:** 4
**Confidence:** 4

**Summary:**

This paper proposes a diffusion-based approach to multi-objective optimization. The core novel mechanism is straightforward: during training, the model learns to map from artificially worsened objectives (real objectives plus a positive shift) to designs, then at inference conditions on actual objectives to encourage improvement. At each reverse diffusion step, they apply post-processing that combines multiple gradient descent and RBF-based repulsion borrowed from existing MOO literature. The method trains a separate model per problem and is evaluated in three settings. The online setting (cheap evaluations) shows marginal gains over simpler gradient methods. The offline setting enables the model to learn from static datasets and propose new designs beyond the dataset. The Bayesian setting trains the conditional diffusion model on small datasets with data augmentation, and demonstrates promising results against some MOBO baselines like PSL-MOBO.

**Strengths:**

- I think the idea of shifting condition is interesting and new in MOO literature.
- The algorithm is straightforward, and the idea of data augmentation that enables the method to work in Bayesian setting is interesting.

**Weaknesses:**

- I find it difficult to grasp the theoretical insight that makes the method work. Here is my understanding: (1) the diffusion model learns the input-output mapping from the training data, (2) it also implicitly learns the global shift vector $\Xi$. Then, if the diffusion model has learned the input-output mapping well and can generalize, it should be able to shift its output by conditioning on the new $C$ to infer $F$ via point 2, and thereby generate $X$ through its generalization capability. This hypothesis could be partially verified by explicitly feeding the shift vector to the conditional diffusion model (i.e., conditioning on both $C$ and $\Xi$) to see if it gives better performance as an ablation study. However, if this is the mechanism, I cannot understand why it would work well in data-sparse settings like the online scenario wher obj is expensive .
- The shift vector introduces a tricky hyperparameter with unclear guidance on how to properly choose it.
- Although benchmarked across different problem settings, the claims seem somewhat overstated. I am not fully convinced this provides significant benefit in the cheap online setting. For the Bayesian setting, it may be beneficial, which is interesting but warrants deeper investigation with more ablation studies since PSL-MOBO is not a very standard baseline. Overall, I think this is more of an algorithm suited for offline settings.
- The method trains a separate model per problem with no mechanism for transfer learning or generalization across problems. This limits scalability and requires expensive retraining (1000 epochs) for each new problem.
- The paper lacks comparison against simpler baselines, particularly: (1) MGD+RBF with multiple random initializations (no diffusion model) in the online setting, (2) guidance-free diffusion sampling to quantify the guidance contribution, and (3) explicit vs implicit shift conditioning. Without these ablations, it is unclear whether the diffusion model provides meaningful benefit beyond the gradient-based guidance.

**Questions:**

- Have the authors compared their approach against MGD+RBF without the diffusion model in all three settings?
- As mentioned in weakness, can the authors provide ablations with explicit shift conditioning $p(x|c,\Xi)$ versus the current implicit approach?

---

> ### Author Response · Authors · 2025-11-21
>
> We thank the reviewer for engaging deeply with our work and offering valuable suggestions to improve the manuscript.
>
> ## **Addressing the Weaknesses (Part I)**
>
> 1. **Clarifying the Core Theoretical Mechanism**
>     - During training, each point $\mathbf{x}$ is paired with the condition $\mathbf{c} = \mathbf{F}(\mathbf{x}) + \Xi,\ \Xi \in (0,\infty)^m.$ Because every coordinate of $\Xi$ is strictly positive, the conditioning value $\mathbf{F}(\mathbf{x})+\Xi$ intentionally represents a vector that exceeds $\mathbf{F}(\mathbf{x})$ in all objectives, and therefore associates each training condition with a region of the objective space that stands strictly "above" the true objective values. The diffusion model is thus trained to generate samples $\mathbf{x}$ whose objective values lie below the shifted condition; more clearly, the model learns to produce samples whose objectives fit underneath the conditioning signal. Theorem 1 formalizes that the learned conditional distribution concentrates on samples that satisfy this dominance property. At sampling time, we condition not on a shifted label but directly on $\mathbf{c}_T = \mathbf{F}(\mathbf{x}_T).$ Consequently, the diffusion model interprets $\mathbf{F}(\mathbf{x}_T)$ as $\mathbf{F}(\mathbf{x})+\Xi$ for some (implicit) $\mathbf{x}$.  The reverse diffusion process therefore produces a refined point $\mathbf{x} = \mathbf{x}_0$ whose objective values tend, by construction, to lie below $\mathbf{F}(\mathbf{x}_T)$, giving a probability of dominance of at least $1-\tau$ (Theorem 1). This explains why conditioning with $\mathbf{F}(\mathbf{x}_T)$ systematically yields improved solutions.
>     - Indeed, diffusion models typically require a sufficiently large training dataset to learn reliable conditional denoising dynamics.  We summarize below how SPREAD accommodates this requirement across all settings considered.
>         - Online Setting (true objectives available). In the online case, we have unrestricted access to the true objective functions. This allows us to generate as many training samples as needed prior to the diffusion training phase. In our experiments, we evaluate the objectives on $10 000$ Latin-hypercube–sampled points, which consistently provides a sufficiently rich dataset for the conditional diffusion model to learn the structural relationship between decision variables and their corresponding shifted objective vectors. Because objective evaluations are inexpensive in this setting, dataset size is not a limiting factor.
>         - Offline Setting (true objectives unavailable). In the offline case, the objective functions are not available during optimization. Instead, each benchmark provides a fixed dataset $\mathcal{D} = {(x, F(x))}$. As shown in Table 6 of the appendix, all offline benchmarks considered in our experiments offer datasets of size $60 000$, which is sufficiently large for training the conditional diffusion model without further augmentation. The diffusion model is therefore trained solely on this static dataset, consistent with the constraints of the offline MOO paradigm.
>         - Bayesian / MOBO Setting (true objectives available but expensive). In the MOBO setting, the true objective functions are available but expensive to evaluate. Since the evaluation budget is small $100$ evaluations in our experiments), directly collecting a large diffusion-training dataset is infeasible. To overcome this limitation, we adopt the data-augmentation strategy of CDM-PSL. This produces a sufficiently large training dataset for the diffusion model, while respecting the evaluation-budget constraints of Bayesian optimization.
>     - *Ablation with explicit shift conditioning.* In Table 11 in the appendix of our revised manuscript, we report the ablation results for explicit shift conditioning. For each training sample $\mathbf{x}^i$, the condition is defined as $c =(\mathbf{F}(\mathbf{x}^i), \Xi)$, instead of the implicit form $c =\mathbf{F}(\mathbf{x}^i)+ \Xi$. This requires embedding $\mathbf{F}(\mathbf{x}^i)$ and $\Xi$ separately and concatenating their embeddings with the time embedding. Consequently, the conditioning input to the multi-head cross-attention module has shape $(n, 3, e)$, while the output retains the shape $(n, 1, e)$ as in Figure 1. During sampling, we set the condition to $c =(\mathbf{F}(\mathbf{x}^i), 0)$.  Overall, the results indicate that the implicit shift conditioning used in SPREAD slightly outperforms explicit shift conditioning in terms of hypervolume on most problems, with the difference being especially notable on the 4-objective RE41 problem. Explicit conditioning, however, improves solution diversity on the ZDT problems and achieves higher hypervolume on RE33 and RE34. Nevertheless, the theoretical improvement guarantee established for implicit shift conditioning in Theorem 1 does not directly carry over to the explicit variant.

---

> > ### Author Response · Authors · 2025-11-21
> >
> > ## **Addressing the Weaknesses (Part II)**
> > - **Choosing the Shift Vector $\Xi$ for the Training Data.** For each training batch $\mathbf{X_b} = \\{\mathbf{x}^i\\}_{i=1}^{N_b}$, we set $\Xi = \omega \cdot \mathbf{1}_m$, where $\mathbf{1}_m$ denotes the $m$-dimensional vector of ones. The value of $\omega$ is determined as
> > $$\omega = \min\\{\mathbf{F}(\mathbf{x}^i) \mid \mathbf{x}^i \in \mathbf{X_b}, \\ \mathbf{F}(\mathbf{x}^i) > 0\\},\\ \text{if this set is nonempty},$$
> > $$\omega = 10^{-6},\\ \text{otherwise}.$$
> >
> > - **Clarifying the Positioning and Applicability of SPREAD.**
> > SPREAD is indeed offline-friendly, as it can leverage large static datasets to generate high-quality Pareto optimal solutions without requiring additional evaluations. However, our results show that it also provides clear advantages in the online regime, even when function evaluations are inexpensive. Specifically, across $11$ online benchmarks, SPREAD consistently matches or outperforms strong gradient-based Pareto set learning methods (PMGDA, MOO-SVGD, HVGrad, and STCH) in terms of hypervolume and $\Delta$-spread (Tables 1 and 2). These methods are designed specifically for online settings where function evaluations are cheap, so outperforming them indicates that SPREAD delivers meaningful benefits beyond what classical online MOO approaches can achieve. Additionally, Figure 2 shows that SPREAD offers a favorable trade-off between computational efficiency and solution quality.
> > Regarding the Bayesian setting, we would like to clarify that our evaluation includes several recent and competitive MOBO baselines beyond PSL-MOBO, such as SVH-PSL (2025), CDM-PSL (2025), PDBO (2024), and qPOTS (2023). These methods span multiple families of approaches (Pareto-set learning, diffusion-based, and acquisition-based), ensuring that our comparisons are broad and representative of the current state of the art.
> >
> > - **Training Cost and Feasibility of Transfer Learning Across Related Problems.**
> > We acknowledge that training a separate diffusion model for each problem introduces additional computational overhead. However, when multiple problems share the same decision space and number of objectives, transfer learning becomes feasible and can substantially reduce the training effort in both online and offline settings. We have added an experiment in which we perform transfer learning by training a single diffusion model on groups of related problems. Two groups are considered in both the online and offline settings:
> >   - Online setting: (ZDT1-3) and (DTLZ2, DTLZ4, DTLZ7)
> >   - Offline setting: (ZDT1-3) and (DTLZ2-7)
> >
> >   Tables 12 and 13 in the appendix present the results. In the online setting, transfer learning achieves performance comparable to training individual models, with only one exception (DTLZ2) where the hypervolume is slightly lower. In the offline setting, transfer learning even improves performance on most problems. Overall, these experiments suggest that transfer learning is a practical strategy for reducing training cost with minimal risk of performance degradation. We sincerely thank the reviewer for highlighting this point.
> >
> > - **MGD+RBF and Guidance Ablation Experiments.**
> > In all three settings, we have included additional experiments using the straightforward MGD+RBF baseline as well as an ablation of SPREAD without any guidance mechanism. For MGD+RBF, we apply the second update of Equation 9 for $T$ iterations (using the same $T$ as in the main experiments) to refine random initializations, without using any diffusion model. For the SPREAD ablation without guidance, we discard the second update of Equation 9 and rely solely on the reverse-diffusion update (the first update of Equation 9). As shown in Table 9 in the appendix, SPREAD consistently dominates both baselines in the online setting. In the offline setting (Table 10), SPREAD(w/o guidance) achieves the best performance on many DTLZ problems, while SPREAD surpasses both baselines on most ZDT and RE problems; MGD+RBF, however, shows strong performance on the RE61 task. In the MOBO setting, Figure 6 demonstrates that SPREAD and SPREAD(w/o guidance) both outperform MGD+RBF on most tasks, with SPREAD performing best overall. These results indicate that both diffusion and guidance are essential components contributing to the strong performance of SPREAD.

---

> > > ### Author Response · Authors · 2025-11-21
> > >
> > > # **Point-by-Point Answers to Reviewer Questions**
> > >
> > > - Yes, we have now added comparisons against the MGD+RBF baseline (without any diffusion model) in all three settings. These new experiments, together with the ablation study of SPREAD without guidance, are presented in the revised appendix (Tables 9 and 10, and Figure 6). We also discuss and interpret these results in the section addressing the reviewer's final weakness.
> > >
> > > - Yes, we have added an ablation comparing explicit shift conditioning with the implicit approach used in SPREAD, conducted in the online setting. The results are provided in the revised appendix (Table 11), and we discuss the findings in our response to the reviewer's first raised weakness.

---

> > > > ### Comment · Reviewer_CrTm · 2025-11-25
> > > >
> > > > I thank the authors for the thorough rebuttal and the efforts on providing new empirical evidence. I think although it is true that in general such framework allows one to generalize to many problem setting, a proper focus of scope would make the claim does not sounds like exaggeration. More specifically,
> > > > 1. the online setting a bit toy as otherwise one should directly using genetic algorithm (without a model).
> > > > 2. I am in a doubt that this will be useful for BO because of the retraining of diffusion model is super expensive.
> > > >
> > > > Given the performance in the offline setting and the interesting approach, I am willing to raise my score to leaninig accept,

---

> > > > > ### Author Response · Authors · 2025-11-25
> > > > >
> > > > > We appreciate Reviewer CrTm's thoughtful reassessment and the constructive perspective on clarifying the scope of our claims. To address this, the discussion of the online and BO settings will be adjusted accordingly. It is encouraging that the reviewer found the offline results and the overall approach compelling, and we are grateful for the improved score.

---

### Official Review · Reviewer_uYzd · 2025-11-01

**Soundness:** 3
**Presentation:** 2
**Contribution:** 3
**Rating:** 6
**Confidence:** 3

**Summary:**

This paper proposes SPREAD, a diffusion-based generative framework for multi-objective optimization (MOO).
The method integrates a conditional DDPM with a Multiple Gradient Descent (MGD)-inspired adaptive guidance mechanism and a Gaussian RBF repulsion term to jointly ensure convergence to Pareto optimality and coverage of the Pareto front.
SPREAD can operate in three major settings:
Online optimization, directly refining decision variables;
Offline optimization, using static datasets via surrogate functions;
Bayesian optimization, using Gaussian process surrogates under tight evaluation budgets.

The paper provides:
- Theoretical results (Theorem 1–2) establishing that the conditional diffusion process yields Pareto-dominant samples with high probability;
- A Transformer-based diffusion architecture (DiT-MOO);
- Extensive experiments across online, offline, and Bayesian benchmarks (ZDT, DTLZ, RE, OffMOO-Bench, and MOBO suites).

Results show that SPREAD outperforms state-of-the-art baselines (PMGDA, HVGrad, STCH, ParetoFlow, PGD-MOO, CDM-PSL) in terms of hypervolume, diversity, and scalability.

**Strengths:**

1. First systematic framework unifying diffusion models and multi-objective optimization.The adaptive gradient guidance that couples DDPM sampling with MGD directions is conceptually novel and practically effective.Theoretical justification (Theorem 1 & 2) provides insight into convergence and dominance guarantees, uncommon in diffusion-based optimization works.
2. Mathematical exposition is rigorous, with precise definitions (Pareto stationarity, dominance, MGD formulation). Algorithmic presentation (Algorithmic 1–4) is detailed enough for reproducibility. Ablation studies (Table 3) isolate the role of each design component (perturbation, repulsion, diversity), showing necessity.Extensive cross-domain experiments strengthen the claim of generality.
3. Addresses three major MOO paradigms (online/offline/Bayesian) under a single framework. Potentially impactful for the community exploring generative optimization or diffusion-driven design. Outperforms strong baselines across different domains and scales; results are consistent and statistically robust.

**Weaknesses:**

1. Theorem 1 assumes total variation (TV) distance ≤ τ between the learned conditional and the true data distribution, but τ is not quantified or empirically verified.
2. Theorem 2 depends on νₜ = 0 for descent guarantees, which does not hold in practice (since diversity requires νₜ > 0). A stability bound for nonzero νₜ would improve rigor.
3. Although Figures 2(a–b) show runtime comparisons, there is no formal analysis of asymptotic computational cost (e.g., O(n·T·d)) or memory scaling.
4. All tested problems have ≤ 4 objectives; no experiments on “many-objective” problems (m ≥ 10).

**Questions:**

1. Theorem 1 assumes that the learned conditional distribution Qθ(⋅∣c) approximates the true conditional training distribution PX∣C=c within TV distance ≤ τ.
  - How is τ estimated or controlled in practice?
  - Can you provide empirical evidence (e.g., KL or Wasserstein estimates) showing that the learned conditional diffusion distribution indeed remains close to the true data distribution during training?
  - If τ > 0.2, how sensitive is Theorem 1’s conclusion to this deviation?
2. Theorem 2 assumes νₜ = 0 for all t, which guarantees a common descent direction.
  - However, experiments rely on νₜ > 0 to ensure diversity.
  - Could you provide an approximate convergence argument or empirical validation that Pareto dominance still holds under small νₜ > 0?
  - How large can νₜ be before the descent guarantee breaks down?
3. The “main direction” optimization mixes alignment and diversity terms and is solved via “a fixed number of gradient steps.”
  - What is the typical number of inner-loop iterations and their computational cost relative to one DDPM denoising step?
  - Does convergence of this inner optimization affect the overall sampling stability?

---

> ### Author Response · Authors · 2025-11-21
>
> We are thankful for the reviewer's careful examination and helpful recommendations.
>
> ## **Addressing the Weaknesses**
>
> 1. **Clarifying the TV-Distance Assumption in Theorem 1.**
> In our setting, the conditioning variables $\mathbf{c}$ are continuous, which makes it infeasible to directly estimate or compute the total variation distance $\tau$ between the learned conditional distribution and the true data distribution. As is standard in the diffusion-model literature, we rely on the assumption that, with sufficient model capacity and training data, the learned conditional distribution closely approximates the data distribution.  Our theoretical result is stated under this widely adopted assumption, which is also reasonable in our setting given the amount of training data and the fact that we use an adaptation of the Diffusion Transformer (DiT), a high-capacity diffusion model.
> To empirically support the dominance claim in Theorem 1, we provide a direct visualization of the initial points and their corresponding sampled points. Since dominance can be visually verified in two objectives, we conduct this experiment on the bi-objective problems considered in the online setting. Figure 4 in the revised appendix presents the results for ZDT1, ZDT2, ZDT3, and RE21. No guidance mechanism is used so that the effect of the diffusion model alone can be clearly assessed. As shown in the plots, the sampled points consistently dominate their initializations, providing empirical validation of the theoretical improvement stated in Theorem 1.
>
> 2. **Theoretical and Empirical Behavior for $\nu_t > 0.$**
> We thank the reviewer for raising this important point. A detailed technical discussion has been added to Appendix A.3 of the revised manuscript. In summary, Theorem 2 analyzes the idealized case $\nu_t = 0$, which guarantees that the update direction coincides with the multi‐objective steepest descent direction. When $\nu_t > 0$, the diversity term induces a deviation between the computed direction $\mathbf{u}^i_t$ and the descent direction $\mathbf{g}^i_t$. In Appendix A.3, we show that descent can still be ensured as long as this deviation remains smaller than the minimal angle between $\mathbf{g}(\mathbf{x})$ and the individual gradients $\nabla f_j(\mathbf{x})$. This provides a condition under which small values of $\nu_t$ preserve Pareto improvement. However, guaranteeing this bound in practice would require costly sample-wise estimation and dynamic adaptation of $\nu_t$, which is impractical.
> Empirically, we find that using a small positive value of $\nu_t$ (we use $\nu_t = 10$ in all experiments) yields Pareto fronts with strong coverage, as illustrated in Figure 7 of Appendix D.
> These results suggest that while the strict theoretical guarantee holds only for $\nu_t = 0$, the method remains stable and effective for the small $\nu_t > 0$ used in our experiments.
>
> 3. **Asymptotic Complexity of the Sampling Procedure.**
> We thank the reviewer for pointing this out. A formal asymptotic analysis of SPREAD's computational and memory requirements is provided in Appendix A.4 of the revised manuscript. In summary, for $n$ points, decision space dimension $d$, number of objectives $m$, $T$ reverse-diffusion steps, and $K$ inner optimization steps, one sampling run has time complexity
> $$
> \mathcal{O}\left(Tnmd + T K (n m d + n^{2} m)\right),
> $$
> dominated by the $\mathcal{O}(T K n^{2} m)$ repulsion term when $n$ is large.
> Memory scales as
> $$
> \mathcal{O}\big(n(d+m) + n^{2} m + n e L + |\theta|\big),
> $$
> where $e$ and $L$ are the hidden dimension and number of DiT-MOO blocks. These results clarify how runtime and memory grow with respect to the key variables $(n, T, m, d)$, and the number of gradient steps in Equation $13$.
>
> 4. **Experiments in the Many-Objective Setting.**
> We have now added experiments on many-objective problems $(m=10)$ by considering the DTLZ problem family, which is flexible with respect to the number of objectives. Since HVGrad is a hypervolume-maximization method, it cannot be applied in this setting because hypervolume computation becomes prohibitively expensive in many-objective optimization, and we were therefore unable to run HVGrad. We therefore exclude it from this experiment. To evaluate performance, we use an approximate hypervolume calculation based on the *hvwfg* (Walking Fish Group algorithm) implementation from the *pygmo* library. As shown in Figure 14, SPREAD outperforms the remaining three baselines on DTLZ4 and DTLZ7, while PMGDA achieves the best performance on DTLZ2. Overall, these results demonstrate that SPREAD remains competitive in the many-objective setting and scales favorably to higher numbers of objectives.

---

> > ### Author Response · Authors · 2025-11-21
> >
> > ## **Point-by-Point Answers to Reviewer Questions**
> >
> > 1. We have addressed this question in our response to the first weakness, where we explain why $ \tau $ cannot be estimated in our continuous-conditioning setting and provide empirical validation of Theorem 1’s dominance claim instead. Regarding the additional point about large deviations (e.g., $\tau > 0.2$), if the approximation error is that high, the reverse-diffusion update alone would not guarantee improvement of the initial points. However, SPREAD incorporates a guidance update that corrects such deviations, ensuring that samples are still steered toward Pareto-improving regions. This robustness is reflected in the empirical results across all settings.
> >
> > 2. This issue is discussed in our response to the corresponding weakness (Point 2) and in Appendix A.3 of the revised manuscript. In summary, we provide a theoretical argument showing that, for sufficiently small $\nu_t > 0$, the deviation between the update direction $\mathbf{u}_t^i$ and the descent direction $\mathbf{g}_t^i$ can be bounded so that descent is preserved. While establishing this bound strictly in practice would require costly sample-wise adaptation of $\nu_t$, we note that all experiments in the paper use a fixed value $\nu_t = 10 > 0$, and the resulting Pareto fronts consistently exhibit strong coverage across benchmarks. This suggests that, although the theoretical guarantee is derived for $\nu_t = 0$, the method remains stable under the small positive $\nu_t$ used in practice.
> >
> > 3. In response to the reviewer’s question, we performed an ablation on the number of main direction optimization steps $(5, 10, 25, 50)$ on ZDT2, DTLZ4, and RE41. The results are shown in Figure 12 in the revised appendix. The runtime plot shows that increasing the number of inner steps increases the cost linearly, but the cost remains modest compared to a full DDPM denoising pass. Regarding performance, the hypervolume and $\Delta$-spread curves do not show a consistent monotonic trend across problems: on RE41, performance improves with more steps, while on DTLZ4, the best values appear at smaller step counts. Importantly, all curves remain within a comparable range across the tested step counts. These observations suggest that using a small fixed number of steps (e.g., $5-25$) is practical and that the method does not appear particularly sensitive to full convergence of the inner update.

---

> ### Comment · Reviewer_uYzd · 2025-11-24
>
> Thank you for your detailed and thorough responses. Based on the revisions, I have raised my score.

---

> > ### Author Response · Authors · 2025-11-25
> >
> > We appreciate Reviewer uYzd's careful reassessment and are glad that the revisions addressed the raised concerns, leading to an improved score.

---

### Author Response · Authors · 2025-11-21

To improve readability, we revised the notation used for the MGD and guidance directions. Since these directions are always computed from $\mathbf{X}'\_{t}$, their dependence on the primed variables is unambiguous, and the prime notation is unnecessary. In the revised manuscript, we therefore replace $\mathbf{g}\_{t}^{'i}$, $\mathbf{h}\_{t}^{'i}$, and $\tilde{\mathbf{h}}\_{t}^{'i}$ with $\mathbf{g}\_{t}^{i}$, $\mathbf{h}\_{t}^{i}$, and $\tilde{\mathbf{h}}\_{t}^{i}$, respectively. All modifications addressing the reviewers' weaknesses and questions have been highlighted in blue throughout the revised manuscript.

---

### Author Response · Authors · 2025-11-29
**Final statement**

Dear editors, dear reviewers,
we would like to emphasize that (even though we obviously cannot prove it) we had no knowledge of the potential leak of reviewer or author names prior to the official mail by the conference organizers. We can thus assure you that the double blind procedure was not violated from our side.

Having said that, we are somewhat shocked that our ratings have been reset. We worked tirelessly for a whole week to address all the reviewers' comments, and improving our scores from 6-6-4 to 8-8-6 was a major success for us, giving us high hopes to actually get accepted at ICLR. In light of these points, we would be very grateful if these improved scores could still be considered, regardless of the leak that was neither our nor the reviewers' fault.

With kind regards
the authors

---

### Meta-Review · Area_Chair_dLVA · 2026-01-07

**Summary:**

This paper proposes SPREAD, a diffusion-based generative framework for multi-objective optimization that learns a conditional Denoising Diffusion Probabilistic Model (DDPM) over decision variables and refines samples during reverse diffusion using an adaptive multiple-gradient-descent-inspired guidance update plus an RBF repulsion term to encourage Pareto convergence and coverage. The paper evaluates SPREAD in three settings (online optimization, offline optimization from static datasets, and MOBO using surrogates) and reports competitive hypervolume and diversity/coverage results across standard suites.

Reviewers’ main concerns were about the realism and rigor of the theoretical assumptions, the role and necessity of diffusion versus simpler gradient-based baselines, computational and scaling costs, hyperparameter sensitivity and defaults, and the scope/positioning of claims across online vs offline vs BO settings. The rebuttal and revised manuscript add multiple targeted ablations and analyses addressing most of these points, and reviewers indicated score increases post-rebuttal.

The revised submission appears to address the main technical and empirical gaps identified in review: it adds critical baselines/ablations isolating diffusion and guidance contributions, clarifies and partially stress-tests the theoretical story, provides scaling and complexity analysis, extends to many-objective experiments, and clarifies hyperparameter defaults. The remaining concerns are primarily about positioning and practical scope (especially for BO and the online setting) and about the unavoidable reliance on theoretical assumptions common in diffusion modeling. Given the strengthened empirical validation and the reported post-rebuttal reviewer movement toward acceptance, I recommend acceptance.

**Reviewer Concerns:**

* Theory assumptions and practical relevance:
  - Authors explain TV distance is not directly estimable in continuous conditioning, keep it as a standard diffusion-model assumption, and add empirical dominance visualizations for the diffusion-only component on bi-objective problems.
  - Authors add a discussion giving a sufficient condition under which small nonzero repulsion still preserves descent, and argue the fixed small value used in experiments behaves stably in practice.
* Computational complexity and scaling:
  - Authors add an asymptotic time/memory complexity analysis and clarify that repulsion can dominate runtime for large sample counts.
  - Authors add an ablation on decision-space dimensionality showing SPREAD remains slower than some light baselines but scales better than PMGDA while maintaining stronger solution quality on the tested setting.
  - Authors add an ablation on the number of inner optimization steps and report limited sensitivity with linear runtime scaling.
* Many-objective experiments:
  - Authors add DTLZ experiments with m >= 10 using approximate hypervolume evaluation; performance is competitive and often strong, though not uniformly best across all DTLZ variants.
* Clarifying what drives performance (diffusion vs guidance vs shift mechanism):
  - Authors add MGD+RBF (no diffusion) baselines and a “no guidance” ablation across settings, showing diffusion and guidance each contribute and that SPREAD generally outperforms both simplified baselines, with some nuanced cases depending on benchmark family.
  - Authors add an explicit shift-conditioning ablation versus the implicit mechanism; results are mixed and the implicit approach tends to be stronger on key cases, with explicit conditioning sometimes improving diversity.
* Hyperparameter clarity and sensitivity:
  - Authors specify default values for key hyperparameters (including those previously unclear), describe how they were chosen, and argue relative ordering vs baselines is stable even when hyperparameters vary widely.
* Scope and positioning:
  - One reviewer remained concerned the online setting may be somewhat toy and that diffusion retraining cost may limit usefulness for BO; authors acknowledge and adjust positioning, and add transfer-learning experiments showing training one model on groups of related problems can match or improve per-problem training in several cases.

**Reviewer Scores:**

Reviewer uYzd (initial 6; raised score post-rebuttal): Likely increases to 7 if full discussion had occurred, since their core concerns (theory caveats, scaling, many-objective experiments, inner-loop cost) were directly addressed with new appendix analyses and experiments, and the reviewer explicitly raised their score after revisions.

Reviewer rKVB (initial 6; raised score post-rebuttal): Likely increases to 7. Their main issues were notation/readability, runtime scaling with decision-space dimension, hyperparameter defaults/sensitivity, and higher-m objectives; authors addressed each with concrete additions and the reviewer indicated concerns were addressed and increased their score.

Reviewer CrTm (initial 4; raised score post-rebuttal): Likely increases to 5. The reviewer’s skepticism centered on mechanism clarity, missing simple baselines/ablations, overstated claims across settings, and lack of transfer; authors added explicit shift ablation, MGD+RBF and guidance-free baselines, transfer learning experiments, and clarified the mechanism and scope. The reviewer noted the offline setting results are compelling and raised their score to leaning accept.

---

### Decision · Program_Chairs · 2026-01-26

Accept (Poster)